RESEARCH CULTURE

# A survey-based analysis of the academic job market

**Abstract** Many postdoctoral researchers apply for faculty positions knowing relatively little about the hiring process or what is needed to secure a job offer. To address this lack of knowledge about the hiring process we conducted a survey of applicants for faculty positions: the survey ran between May 2018 and May 2019, and received 317 responses. We analyzed the responses to explore the interplay between various scholarly metrics and hiring outcomes. We concluded that, above a certain threshold, the benchmarks traditionally used to measure research success – including funding, number of publications or journals published in – were unable to completely differentiate applicants with and without job offers. Respondents also reported that the hiring process was unnecessarily stressful, time-consuming, and lacking in feedback, irrespective of outcome. Our findings suggest that there is considerable scope to improve the transparency of the hiring process.

**JASON D FERNANDES[†], SARVENAZ SARABIPOUR[†], CHRISTOPHER T SMITH, NATALIE M NIEMI, NAFISA M JADAVJI, ARIANGELA J KOZIK, ALEX S HOLEHOUSE, VIKAS PEJAVER, ORSOLYA SYMMONS[‡], ALEXANDRE W BISSON FILHO AND AMANDA HAAGE***

**\*For correspondence:** amanda. haage@und.edu

[†]These authors contributed equally to this work

**Present address:** [‡]Max Planck Institute for Biology of Ageing, Cologne, Germany

**Competing interests:** The authors declare that no competing interests exist.

## Introduction

The number of PhDs awarded in science, technology, engineering and mathematics (STEM) has increased dramatically over the past three decades (*Cyranoski et al., 2011*; *Ghaffarzadegan et al., 2015*), but the number of faculty positions available has essentially remained constant (*Schillebeeckx et al., 2013*). In the US, for instance, the situation has not changed significantly since 2003, when the National Institutes of Health (NIH) received a major budget increase (*Alberts et al., 2014*). Given the low numbers of faculty positions compared to the numbers of PhDs produced (*Larson et al., 2014*; *Committee to Review the State of Postdoctoral Experience in Scientists and Engineers, 2014*), trainees are limited in their job prospects. Many also emerge from academic training feeling underprepared and under-mentored for any other type of job search (*McDowell et al., 2015*). This leads to a high number of applicants per academic position, many of whom are uncertain about their chances of obtaining a faculty job (*Grinstein and Treister, 2018*; *Sauermann and Roach, 2016*).

Cohorts of new PhDs are also both more diverse than before and more diverse than many current hiring committees (*Alberts et al., 2014*; *White, 2019*; *Bhalla, 2019*). Scientific publishing is also faster-paced than it used to be: for example, evolutionary biologists recruited as "junior researchers" in 2013 had published nearly twice as many articles (22 ± 3.4) as those hired in 2005 (12.5 ± 2.4); the same study also found that the length of time between first publication and recruitment as a faculty member had increased from 3.25 (±0.6) to 8.0 (±1.7) years (*Brischoux and Angelier, 2015*). Longer training periods have been reported repeatedly in many STEM fields, and are perceived as detrimental to both the greater scientific community and individuals in temporary postdoctoral positions (*Committee to Review the State of Postdoctoral Experience in Scientists and Engineers, 2014*; *Ahmed, 2019*; *Rockey, 2012*; *Acton et al., 2019*).

Despite these changes, the academic job search has largely remained the same, resulting in academic hiring being perceived as an opaque process with no clear standards or

guidelines. Beyond a requirement for a doctoral degree and possibly postdoctoral training, faculty job advertisements rarely contain specific preferred qualifications. Furthermore, the criteria used to evaluate applicants are typically determined by a small departmental or institutional committee and are neither transparent nor made public. The amount of materials required for faculty job applications is also highly variable among hiring institutions, and often places a heavy burden on both applicants and search committees (*Lee, 2014*).

Previous studies agree on a need to increase transparency in career outcomes and hiring practices (*Golde, 2019*; *Polka et al., 2015*; *Wright and Vanderford, 2017*). The annual pool of faculty job applicants is large and provides a unique opportunity for examining the application process. We performed an anonymous survey, asking applicants for both common components of research and scholarly activity found on an academic CV, as well as information on their success through the 2018–2019 job cycle. We further performed a small-scale, complementary survey of search committee members. Here we present qualitative and quantitative data on the academic job market, including information on the number of successful off-site and on-site interviews, offers, rejections, and the lack of feedback.

Job applicants start by searching for relevant job postings on a variety of platforms (*Supplementary file 1*). The initial electronic application generally consists of a cover letter addressing the search committee, a teaching philosophy statement, CV, and a research plan (*Figure 1*). The length and content of these materials can vary drastically based on the application cycle, region, institution, or particular search committee. In the current system, the expectation is that application materials be tailored for each specific institution and/or department to which the applicant is applying. This includes department-specific cover letters (*Fox, 2018a*), but may also involve a range of changes to the research, teaching, and diversity statements.

The search committee convenes for a few meetings to shortlist the applicants. Applicants are then contacted for interviews somewhere between one to six months after application materials are due. Searches may include an initial off-site (remote) interview, followed by an on-site interview at the hiring university. The on-site interview typically lasts one or two days and consists of a research seminar, possibly a teaching demonstration, and likely a chalk-talk (*Rowland, 2016*). The on-site interview also usually consists of one-on-one meetings with other faculty members, including a meeting with the hiring department chair, trainees, and the administrative staff.

After the interviews, candidates may be contacted and offered a position, usually in writing. The offer package will include the proposed start date, salary and start-up funds (*Macdonald, 2019*). The time to offer is also variable, but is usually shorter than the time between application and first contact (based on anecdotal information). Importantly, a single search can result in multiple offers (for instance the department may be able to fund multiple competitive candidates, or the first-choice candidate may decline and the second candidate is given an offer). Searches can also fail if the committee does not find a suitable candidate for their program/department or "go dry" if the applicant(s) deemed qualified by the search committee decline their offer.

## Results

We designed a survey for early-career researchers aimed at bringing transparency to the academic job market (see Materials and methods and *Supplementary file 41*). The survey was distributed via Twitter, the Future PI Slack group, and email listservs of multiple postdoctoral associations, resulting in 322 responses from self-identified early-career researchers who applied for academic positions in the 2018–2019 application cycle. Of these, data from 317 respondents passed simple quality filters and were used for analyses. As all questions were optional, these 317 responses represent the maximum number in our analyses; in cases where respondents chose not to answer the question, we analyzed only the applicant subset with responses and list the number of responses used for each analysis in the appropriate figures and supplementary files.

### Demographics of respondents

Respondents reported a large range in the number of submitted applications from a minimum of one to a maximum of 250 (median: 15). The respondent pool was notably enriched in

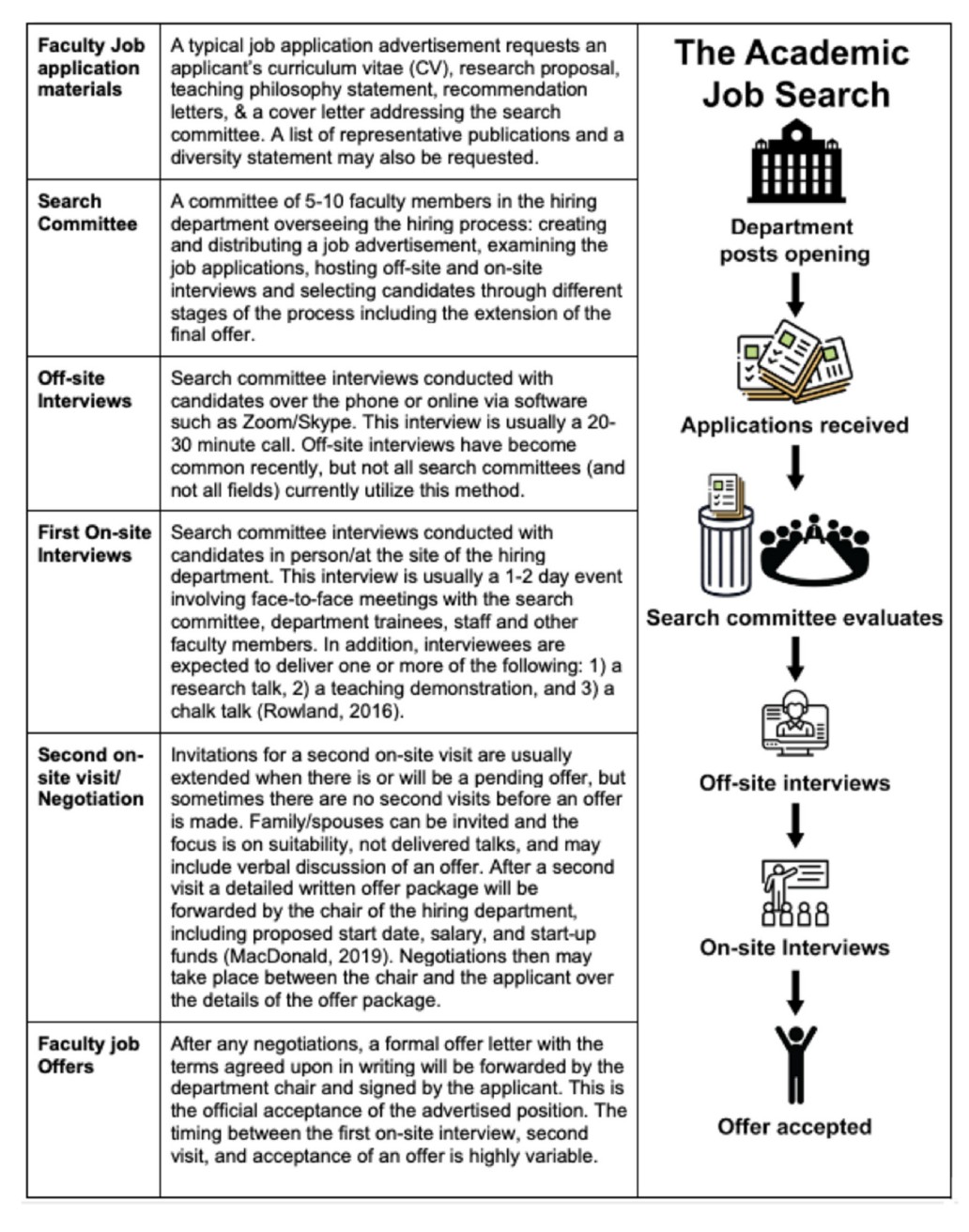

| | |
|---|---|
| **Faculty Job application materials** | A typical job application advertisement requests an applicant's curriculum vitae (CV), research proposal, teaching philosophy statement, recommendation letters, & a cover letter addressing the search committee. A list of representative publications and a diversity statement may also be requested. |
| **Search Committee** | A committee of 5-10 faculty members in the hiring department overseeing the hiring process: creating and distributing a job advertisement, examining the job applications, hosting off-site and on-site interviews and selecting candidates through different stages of the process including the extension of the final offer. |
| **Off-site Interviews** | Search committee interviews conducted with candidates over the phone or online via software such as Zoom/Skype. This interview is usually a 20-30 minute call. Off-site interviews have become common recently, but not all search committees (and not all fields) currently utilize this method. |
| **First On-site Interviews** | Search committee interviews conducted with candidates in person/at the site of the hiring department. This interview is usually a 1-2 day event involving face-to-face meetings with the search committee, department trainees, staff and other faculty members. In addition, interviewees are expected to deliver one or more of the following: 1) a research talk, 2) a teaching demonstration, and 3) a chalk talk (Rowland, 2016). |
| **Second on-site visit/ Negotiation** | Invitations for a second on-site visit are usually extended when there is or will be a pending offer, but sometimes there are no second visits before an offer is made. Family/spouses can be invited and the focus is on suitability, not delivered talks, and may include verbal discussion of an offer. After a second visit a detailed written offer package will be forwarded by the chair of the hiring department, including proposed start date, salary, and start-up funds (MacDonald, 2019). Negotiations then may take place between the chair and the applicant over the details of the offer package. |
| **Faculty job Offers** | After any negotiations, a formal offer letter with the terms agreed upon in writing will be forwarded by the department chair and signed by the applicant. This is the official acceptance of the advertised position. The timing between the first on-site interview, second visit, and acceptance of an offer is highly variable. |

**Figure 1.** An overview of the academic job search process. The first column defines common terms in the academic job search; while the second column outlines how the search for an academic job progresses, from a job being posted to an offer being accepted.

applicants who received at least one off-site interview (70%), at least one on-site interview (78%) and at least one offer (58%); this may represent a significant bias towards successful applicants in our study, as a recent study shows that less than 23% of PhDs eventually secure a tenure-track position (*Langin, 2019*).

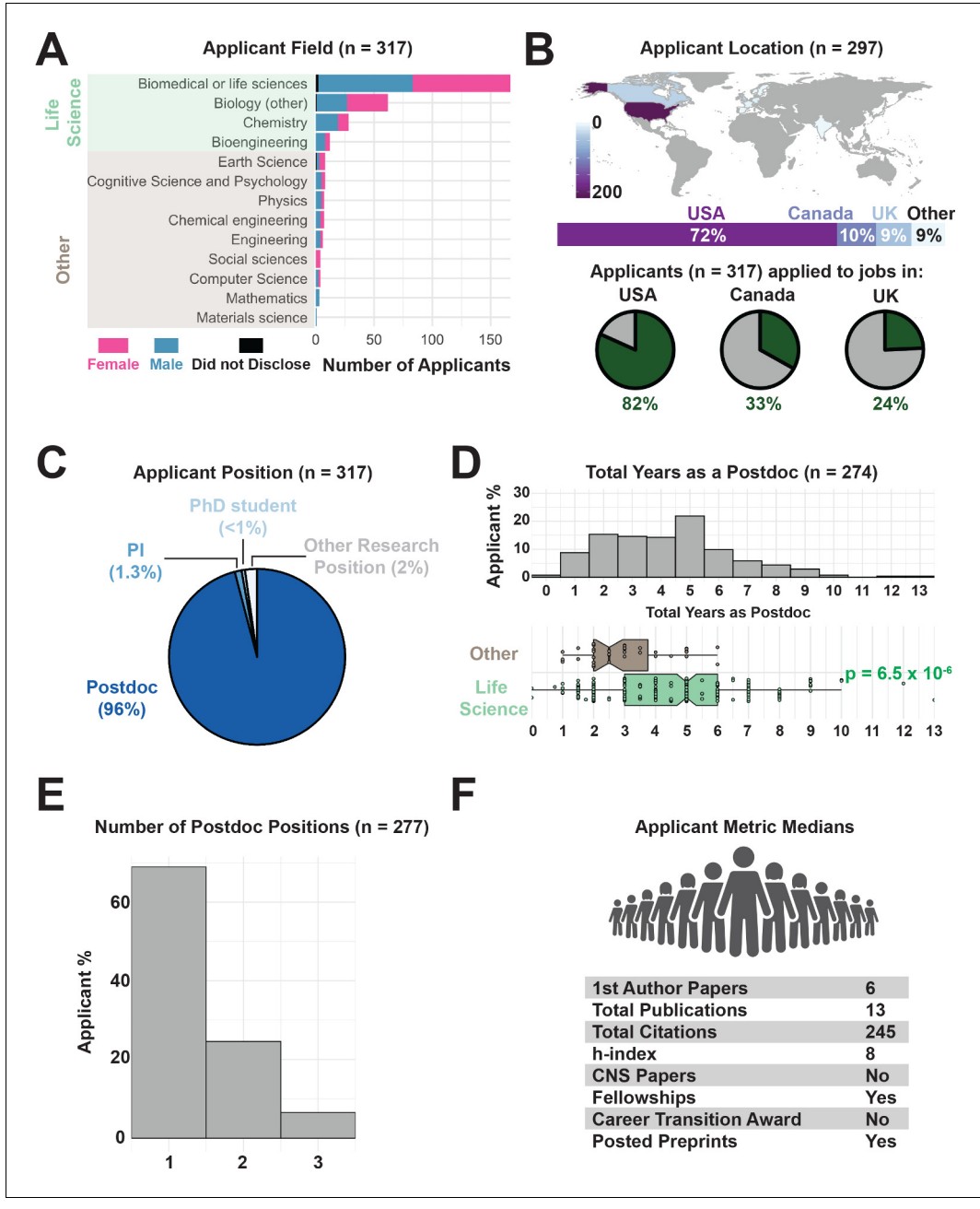

**Figure 2.** Demographics of academic job applicants. (A) Distribution of survey respondents by self-identified gender and scientific field (*Supplementary file 2*). Fields highlighted in green were grouped together as life-science related fields for subsequent analyses. (B) Distribution of countries where respondents were researching at the time of the survey (top, see *Supplementary file 3*) and the countries in which they applied to faculty jobs (green slices of pie charts, bottom; see *Supplementary file 4*). (C) Self-reported positions of applicants when applying for faculty jobs (*Supplementary file 5*). (D) The number of years spent as a postdoctoral researcher ranges from 1 year or fewer (4% of applicants) to eight or more years (9% of applicants; maximum of 13 years, top). Life-science related postdoctoral training (n = 268 respondents) takes significantly longer than in other fields (n = 49 respondents; p=6.5×10$^{-6}$, bottom; for data see *Supplementary file 6*; for statistical analysis see *Supplementary file 7*). (E) Number of postdoctoral positions held by survey applicants (*Supplementary file 8*). (F) Median values for metrics of research productivity in the applicant pool (*Supplementary file 9*).

Respondents represented researchers in a wide variety of fields, with 85% from life sciences and related fields, with relatively equal numbers of applications from men and women across this group (*Figure 2A*). Our survey captured data from an international applicant pool, representing 13 countries (*Figure 2B*). However, 72% of our respondents reported currently working in the United States, which may reflect the larger circulation of our survey on social media platforms and postdoctoral associations there. Most candidates applied to jobs within the United States (82%), Canada (33%), and the United Kingdom (24%). 96% of respondents entered the job market as postdoctoral researchers (*Figure 2C*). The applicants spent 1 to 13 years (median: 4 years) in a postdoctoral position. These data are consistent with a recent report suggesting that postdocs in the United States across a variety of fields spend an average of 2.5–3.6 years in their positions (*Andalib et al., 2018*).

Notably, in our survey population, postdocs in the life sciences spent a median of 5 years in a postdoctoral position, significantly longer than those in other fields, who reported a median postdoc length of 2.75 years prior to applying for a faculty position (*Figure 2D*), consistent with previous findings on increased training times in the life/biomedical sciences before junior faculty recruitment (*Committee to Review the State of Postdoctoral Experience in Scientists and Engineers, 2014*; *Brischoux and Angelier, 2015*; *Ahmed, 2019*; *Powell, 2017*; *Rockey, 2012*). 68% of respondents went on the job market while in their first postdoctoral position (*Figure 2E*).

Applicants had a large range in their publication records, including number of papers co-authored, h-index, and total citation count. Respondents reported a median of 13 total publications (including co-authorships and lead authorships), with a median of 6 first author papers when entering the job market (*Figure 2F*).

### Publishing metrics by gender

Gender bias in publishing and evaluation is well documented (*Aileen Day and Boyle, 2019*; *Centra and Gaubatz, 2000*; *Cameron et al., 2016*; *Witteman et al., 2019*). The respondents to our survey were relatively evenly distributed across self-identified genders, with 51% identifying as male, 48% as female, and 1% preferring not to disclose this information (no applicants identified as non-binary; *Figure 3A*). Men reported significantly more first-author publications, total publications, overall citations, and a higher h-index compared to women (*Figure 3B*); more men also reported being authors on papers in three journals with high impact factors (Cell, Nature and Science; *Figure 3C*) than women. The gender differences we observe mirror those seen in other reports on differences in citation counts in STEM fields based on the corresponding author gender (*Schiermeier, 2019*). Despite popular discussions on a need for papers in Cell, Nature, Science or other journals with a high impact factor (*Brock, 2019*; *McKiernan et al., 2019*), 74% of respondents were not authors on a paper in Cell, Nature or Science (CNS), and a greater majority (~84%) did not have a first author publication in these journals (*Figure 3C*). Of the 51 respondents with papers in these journals, 49 (96%) were in a life science-related field, indicating that the valuation of these journals was highly field-specific (*Figure 3C*).

While 78% of respondents reported having obtained fellowships at some point in their career, this figure was 87% for women and 72% for men (*Figure 3D*). Women had better success at receiving both doctoral and postdoctoral fellowships. However, the questions in our survey did not distinguish between the types (e.g. government funded versus privately funded, full versus partial salary support) or number of fellowships applied to; many of these factors are likely critical in better understanding gender differences in fellowship support (*Figure 3D*).

### Applications, interviews and offers

The 317 respondents submitted a total of 7644 job applications in the 2018–2019 application cycle, with a median of 15 applications per respondent (*Figure 4A*). Applicants were invited for a total of 805 off-site interviews (phone, Zoom or Skype; median: 1) and 832 onsite or campus interviews (median: 2), receiving 359 offers (median: 1; *Figure 4A*). Although many hiring processes consist of an off-site (remote) interview, we found that this was not standard since the typical applicant received more on-site than off-site interviews. In our dataset, 42% of participants received no offers, 33% received one offer, 14% received two offers, 6% received three offers, and 6% received more than three offers. Candidates who received offers typically submitted more applications than those who received no offers, indicating that some candidates may not have submitted enough applications to have a reasonable chance of getting an

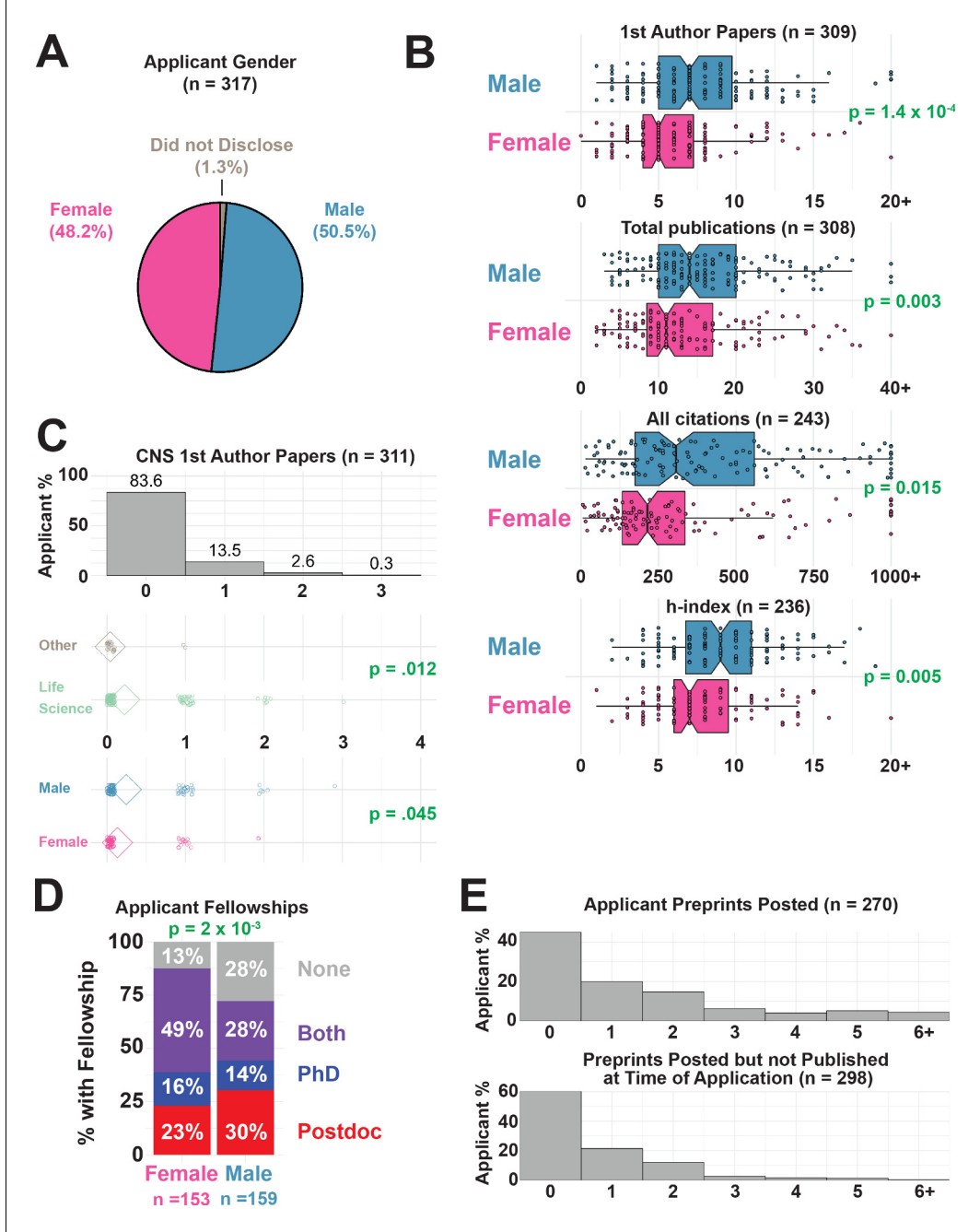

**Figure 3.** Applicant scholarly metrics by gender. (**A**) Distribution of gender (male, female, did not disclose) amongst survey respondents (***Supplementary file 2***, first row). (**B**) Publication metrics of survey respondents including number of first author papers (top), total publications (middle top), total citations (middle bottom), and h-index (bottom) for male and female respondents. Men in our survey reported more first-authored papers than women (medians of 7 and 5, respectively; p=$1.4\times10^{-4}$), more total publications (medians of 16 and 11; p=$3.0\times10^{-3}$), more overall citations (medians of 343 and 228; p=$1.5\times10^{-2}$), and a statistically significant higher h-index (medians of 9.0 and 7.0; p=$5.40\times10^{-3}$; see ***Supplementary files 7*** and ***9***). (**C**) Although most applicants (83.6%) did not have first-author papers in CNS, those in the life sciences had more than applicants in other fields (p=0.012), and men had more than women (p=0.45; see ***Supplementary files 7*** and ***11***). Note: CNS papers do not include papers in spin-off journals from Cell, Nature or Science. (**D**) Distribution of funding reported within training period (doctoral fellowship only in blue, postdoctoral fellowship only in red, fellowships during PhD and postdoc in purple, and no fellowship in gray). Females reported significantly more fellowship funding than males (42% of women vs 36% of men for predoctoral fellowships, and 72% of women, 58% of men for postdoctoral fellowships,

*Figure 3 continued on next page*

*Figure 3 continued*

p=2.40×10$^{-3}$, $\chi^2$ = 12.10, Chi-squared test, df = 2, see *Supplementary files 7* and *13*). (E) Preprints were posted by 148 of 270 (55%) individual candidates, with an average of 1.57 preprints reported per candidate (top). Number of preprints posted which were not yet accepted for journal publication (bottom) while applying for faculty jobs (see *Supplementary file 14*).

offer (*Figure 4A,D*). According to a recent poll on Twitter (which received over 700 responses), most faculty received between one and three offers when they were applying for faculty positions (*Whitehead, 2019*; *Supplementary file 15*).

Despite the fact that successful candidates submitted more applications, the number of applications per candidate did not correlate with the number of offers, while being only weakly correlated with the number of off-site interviews (*Figure 4B*). Not surprisingly, the number of on-site interviews strongly correlated with the number of offers received (*Figure 4C*, bottom). Population medians changed slightly by gender as men submitted slightly more applications, but received slightly fewer off-site interviews. These small differences by gender were not statistically significant (*Figure 4A*). The median number of offers also did not vary by gender.

We split our population into two groups by application number, one group either at or below the median ($\leq$15 applications, n = 162) and the other group above the median (>15 applications, n = 155). These groups had a significant difference in success rates: respondents who submitted more than 15 applications had a significantly higher average number of off-site interviews (*Figure 4D*). We also asked whether respondents applied for non-faculty positions during this cycle (*Supplementary file 16*). 71% of applicants did not apply for other jobs and these applicants had a small, but significant increase in offer percentage (*Figure 4E*).

Taken together, these data seemingly indicate that increasing the number of applications submitted can lead to more interviews, as suggested by others (*Jay et al., 2019*), with the typical candidate submitting at least 15 applications to achieve one offer. However, the lower correlation between application number and offers (compared to application number and interviews) suggests that while higher application numbers can generate more interview opportunities, other criteria (e.g. the strength of the interview) are important in turning an interview into an offer.

## Publication related metrics

The number of papers published, and the impact factors of the journals these papers were published in, can influence the chances of an early-career researcher obtaining an independent position (*van Dijk et al., 2014*; *Powdthavee et al., 2018*). As mentioned previously, it is widely believed that you need a paper in Cell, Nature or Science to secure a faculty position in the life sciences (*McKiernan et al., 2019*; *Sheltzer and Smith, 2014*; *Fox, 2018b*). Our data demonstrates that a CNS paper is not essential to an applicant receiving a faculty job offer.

The majority (74%) of our respondents were not an author on a CNS paper (*Figure 5A*), and yet most participants received at least one offer (58%). However, applicants with a CNS paper did have a higher number of onsite interviews and faculty job offer percentage. Of our respondents, 16% were first author on a CNS paper, and these applicants had a significantly higher percentage of offers per application (p=1.50×10$^{-4}$, median offer percentages: 11% with a CNS paper and 2% without a CNS paper) and on-site interviews (p=2.70×10$^{-4}$, median onsite interview percentages: 21% with a CNS paper, and 10% without a CNS paper; *Figure 5A*).

Since the number of on-site interviews and offers are highly correlated (*Figure 4C*), it is unclear if this increased success simply represents a higher chance at landing more onsite interviews. It is important to note that this effect is correlative and these candidates likely had other attributes that made them appealing to the search committee(s).

We examined several other publication metrics and found no correlation with the number of offers. Specifically, the total number of publications, the number of first author publications, the number of corresponding author publications, and h-index did not significantly correlate with offer percentage (*Figure 4—figure supplement 1*). When we separated candidates who were above and below the medians for each of these metrics and compared the distribution of offer percentages, only the total number of citations significantly associated with a higher offer

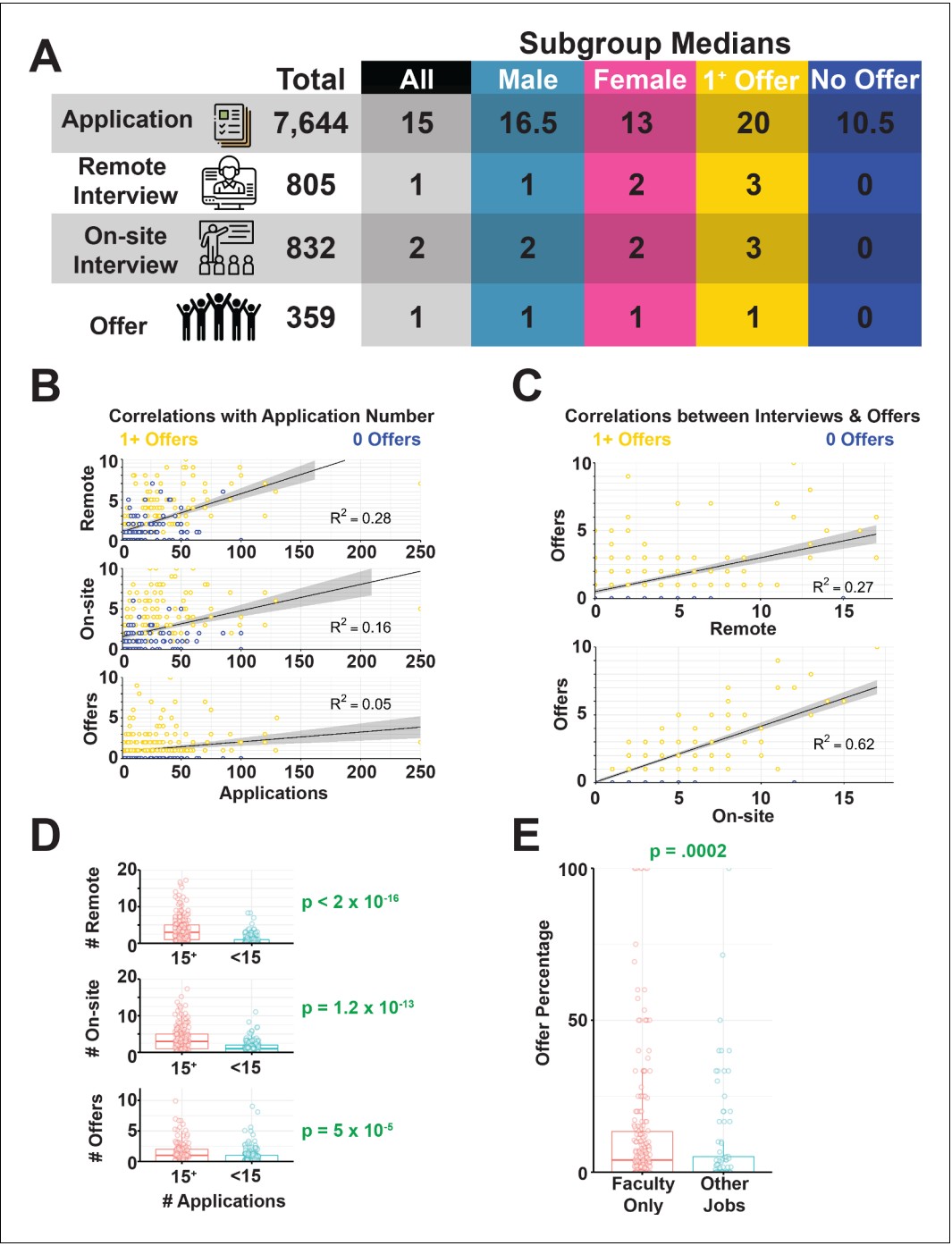

**Figure 4.** Job application benchmarks and their impact on success. (A) Total and median numbers of applications, off-site interviews, on-site interviews and offers recorded in survey responses (***Supplementary file 19***). (B) Correlations between the total number of applications submitted and off-site interviews (top; $R^2 = 0.28$), onsite interviews (middle) and offers (bottom; $R^2 = 4.77 \times 10^{-2}$). (C) Correlations between the number of interviews completed and offers received ($R^2 = 0.62$). See ***Figure 4—figure supplement 1*** for more details. (D) Total number of off-site interviews (top, $p < 4.10 \times 10^{-24}$, on-site interviews (middle, $p = 1.20 \times 10^{-13}$) and offers (bottom, $p = 5.0 \times 10^{-5}$) for applicants who submitted at least 15 (the median) applications (in red) and less than 15 applications (in blue). (E) Fraction of applications that resulted in offers (offer percentages) for survey respondents who did not apply for jobs outside of faculty positions is significantly higher ($p = 2.0 \times 10^{-3}$, ***Supplementary file 7***) than for those who also applied for both academic and other types of jobs (***Supplementary file 14***).

The online version of this article includes the following figure supplement(s) for figure 4:

*Figure 4 continued on next page*

**Figure supplement 1.** Correlations between offer percentage and a number of traditional scholarly metrics.

percentage (*Figure 5B*). Although the offer percentage was generally higher for applicants above the median for the other metrics, none of these differences were statistically significant (*Figure 5B*).

### Preprints

Preprints, or manuscripts submitted to an open-access server prior to peer-reviewed publication, are becoming increasingly popular among early-career researchers (*Sever et al., 2019*), particularly in the life sciences, and can boost article citations and mentions (*Sarabipour et al., 2019*; *Fraser et al., 2019*; *Abdill and Blekhman, 2019*; *Conroy, 2019*; *Fu and Hughey, 2019*).

We received 270 applicant responses on the use of preprints; 55% of respondents had posted at least one preprint, and 20% had posted between two and six preprints (*Figure 3E*, top). At the time of faculty job application, 40% of these respondents had an active preprint that was not yet published in a journal (*Figure 3E*, bottom), with an average of 0.69 active preprints per person. A number of candidates commented that preprinted research was enormously helpful and served to demonstrate productivity before their paper was published (*Supplementary files 17* and *18*).

### Fellowships and career transition awards

Respondents were highly successful in obtaining fellowship funding during their training (80% received a fellowship of any kind, *Figure 3D*). Applicants with a postdoctoral fellowship had a greater offer percentage than those without, although the effect was not significant after correcting for multiple comparisons (p=0.17); doctoral fellowships did not appear to influence offer percentage (*Figure 5B*).

Receiving funding as an early-career researcher is part of a favorable research track record (*Eastlack, 2017*). A recent study of publicly available data indicates that the proportion of faculty receiving their first large research program grant (an R01 through the NIH) with a history of funding as a trainee (F and K awards through the NIH) is significantly increasing, driven mostly by K awards. Pickett states: "While not a prerequisite, a clear shift is underway that favors biomedical faculty candidates with at least one prior training award" (*Pickett, 2019*).

Our survey differentiated the types of funding a trainee can receive into predoctoral and postdoctoral fellowships (discussed above), and career transition awards, for which the trainee is listed as the PI and funds can often transition with the trainee to a hiring institute (e.g. the Burroughs Wellcome Fund Career Awards at the Scientific Interface or the NIH K99/R00 Pathway to Independence award). Career transition awards were less frequent, with 25% of respondents receiving awards on which they were PI/co-PI (*Supplementary file 20*). Respondents with transition funding received a higher percentage of offers (*Figure 5B*).

### Patents

Patents are considered positive metrics of research track record, although their importance and frequency can vary between fields. Only 19% of applicants reported having one or more patents on file from their work when entering the job market (*Supplementary file 21*). The number of patents held by the applicant did not correlate with the number of offers received (*Figure 4—figure supplement 1*) and the percentage of offers did not change between those with or without a patent (*Figure 5B*).

### Years on the job market

We also asked how many application cycles they had been involved in. Approximately 55% of our respondents were applying for the first time, and these candidates fared significantly better in terms of offer percentages than those who were applying again (*Figure 5B*). Additionally, a number of applicants took advantage of resources that provided information about the job application process (*Supplementary file 22*), and those that did found them helpful (*Supplementary file 23*).

Analyses such as the work presented here may help applicants refine and present their materials and track record in a manner that might improve success and decrease repeated failed cycles for applicants.

### Interplay between metrics

We next examined the relationship between each of the traditional criteria that were significantly associated with an increase in offer percentage. The criteria included being first author

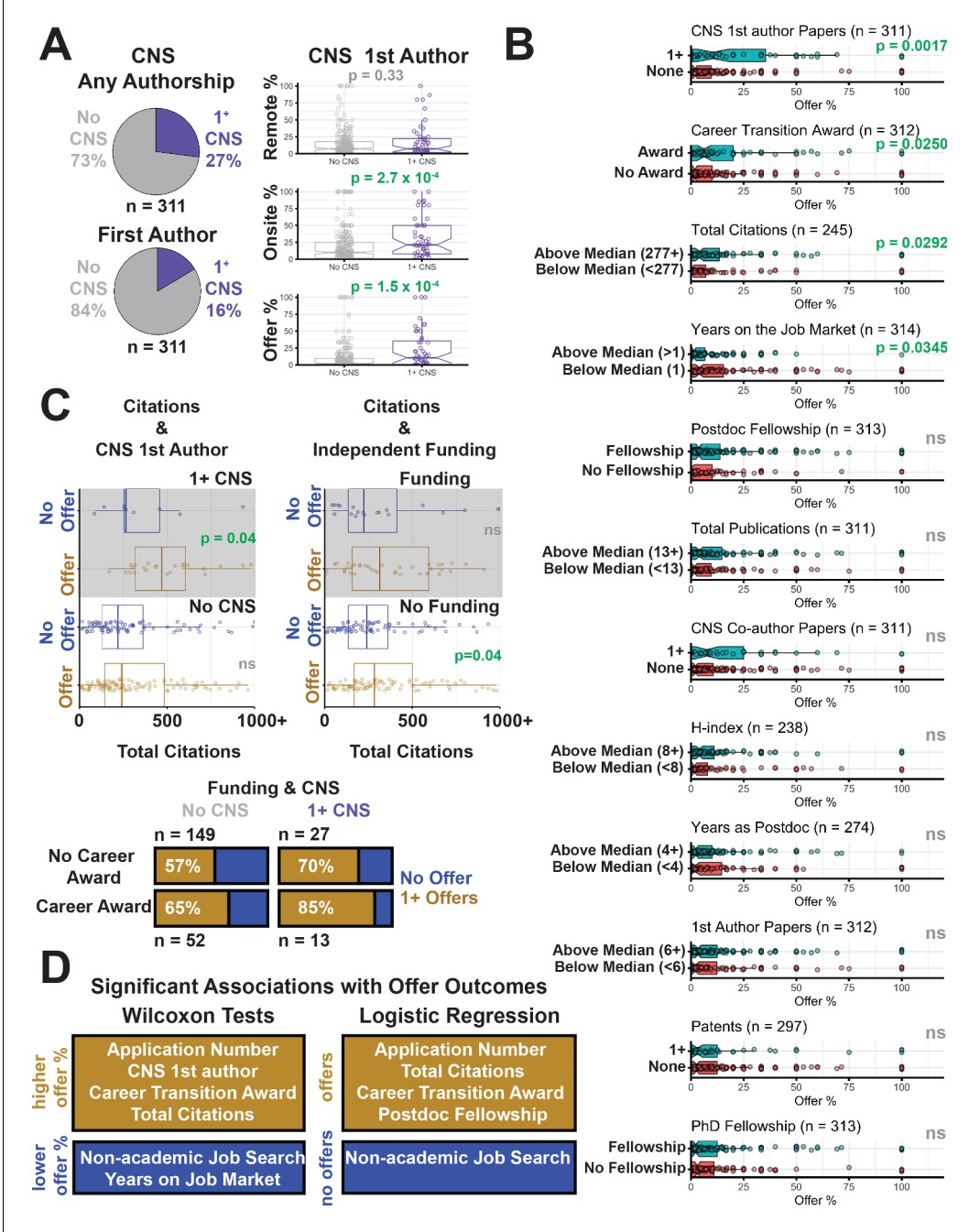

**Figure 5.** Traditional research track record metrics slightly impact job search success. (**A**) Pie charts show the fraction of candidates with authorship of any kind on a CNS paper (purple) versus those without (gray), and fraction of candidates who were first author on a CNS paper (purple) versus those who were not (gray). Distributions of off-site interviews (top; p=0.33), onsite interviews (middle; p=2.70×10⁻⁴) and offers (bottom; p=1.50×10⁻⁴) for applicants without a first-author paper in CNS (gray), and those with one or more first-author papers in CNS (purple; *Supplementary files 11*, *12*, *17*). (**B**) Significant associations were found between offer percentage and the number of first-author papers in CNS (top panel, p=1.70×10⁻³), career transition awards (second panel, p=2.50×10⁻²), total citations (third panel, p=2.92×10⁻²), and years on the job market (fourth panel, p=3.45×10⁻²). No significant associations were found between offer percentage and having a postdoc fellowship (fifth panel), being above the median in the total number of publications (sixth panel), being an author in any position on a CNS paper (seventh panel), h-index (eighth panel), years as a postdoc (ninth panel), number of first-author papers (tenth panel), number of patents (eleventh panel), or graduate school fellowship status (twelfth panel; *Supplementary files 6*, *7*, *9*, *10*, *11*, *12*, *13* and *21*). (**C**) The plots show total citations for those without an

*Figure 5 continued on next page*

*Figure 5 continued*

offer (blue) and those with one or more offers (gold), for all applicants with one or more first-author papers in CNS (top left); for all applicants without a first-author paper on CNS (bottom left); for all applicants with independent funding (top right); and for all applicants without independent funding (bottom right). In two cases the p value is below 0.05. The bar charts show the offer percentages (gold) for the four possible combinations of career award (yes or no) and first-author paper in CNS (yes or no): for applicants with a first-author paper in CNS, p=0.56, $\chi^2 = 0.34$; for applications without, p=0.17, $\chi^2 = 1.92$. (D) Summary of significant results testing criteria associated with offer outcomes through Wilcoxon analyses (***Supplementary file 7***) or logistic regression (***Supplementary file 24***).

The online version of this article includes the following figure supplement(s) for figure 5:

**Figure supplement 1.** Life-science specific analysis of applicant survey outcomes.
**Figure supplement 2.** Visualization of possible paths to an offer using the C5.0 decision tree algorithm.

on a CNS paper, total citations, and career transition awards.

Overall, we had 241 applicants that fully responded to all of our questions about these metrics. Pairwise testing of each of these criteria found no statistically significant relationships between variables (p=0.45, career transition awards vs CNS; p=0.26 total citations vs CNS; p=0.29 career transition awards versus total citations). Regardless, we plotted subgroups based on offer status and each of these criteria to see if there was evidence for any general trends in our dataset (***Figure 5C***). Notably, respondents who were first author on a CNS paper and received at least one offer had a greater number of total citations than those who were first author on a CNS paper but did not receive any job offers. Applicants who were first author on a CNS paper or who had a career transition award had higher percentages of securing at least one offer, and those with both had an even greater percentage, although the differences between these groups was not statistically significant.

This analysis suggests that the combination of different criteria holistically influence the ability to obtain an offer. Therefore, we performed logistic regression to examine the relationship between multiple variables/metrics on the successful application outcome of receiving an offer on a subset of applicants (n = 105) who provided answers across all variables. We implemented a rigorous variable selection procedure to maximize accuracy and remove highly correlated variables. This resulted in a model that included only seven variables (***Supplementary file 24***).

This regression model revealed that a higher number of applications, a higher citation count and obtaining a postdoctoral fellowship were significantly associated with receipt of an offer. When missing values were imputed and the full applicant pool (n = 317) was considered, all previous variables remained significant, and a significant positive coefficient was also observed for having a career transition award. In both versions of the model, the search for non-academic jobs was significantly negatively associated with offer status (***Figure 5D***). We note that the model with imputed data was more accurate than that with missing values excluded at distinguishing between applicants with and without offers in 10-fold cross-validation experiments. However this accuracy was found to only be 69.6%, which is insufficient to construct a usable classifier of offer status. Due to the predominance of applicants from the life sciences in our dataset, we also repeated these analyses on a subset containing only these applicants. While more variables were included in the model, the general trends remained the same, with the addition of the number of years spent on the job market as a significant negative factor in receiving an offer (***Figure 5—figure supplement 1***; ***Supplementary file 25***).

Finally, we extended this analysis to visualize the interplay between all variables in ***Figure 5B*** by learning a decision tree automatically from the collected data (***Figure 5—figure supplement 2***). The algorithm tries to partition the applicants into groups such that each group is entirely composed of individuals with at least one offer or without. A variety of different classifier groups were identified, but no group contained more than ~19% (61 out of 317) of the dataset. In fact, the accuracy of the overall decision tree in distinguishing between candidates with offers and those without was only ~59% (***Figure 5—figure supplement 2***).

Taken together, these results suggest that there are multiple paths to an offer and that the variables we collected do not sufficiently capture this variability.

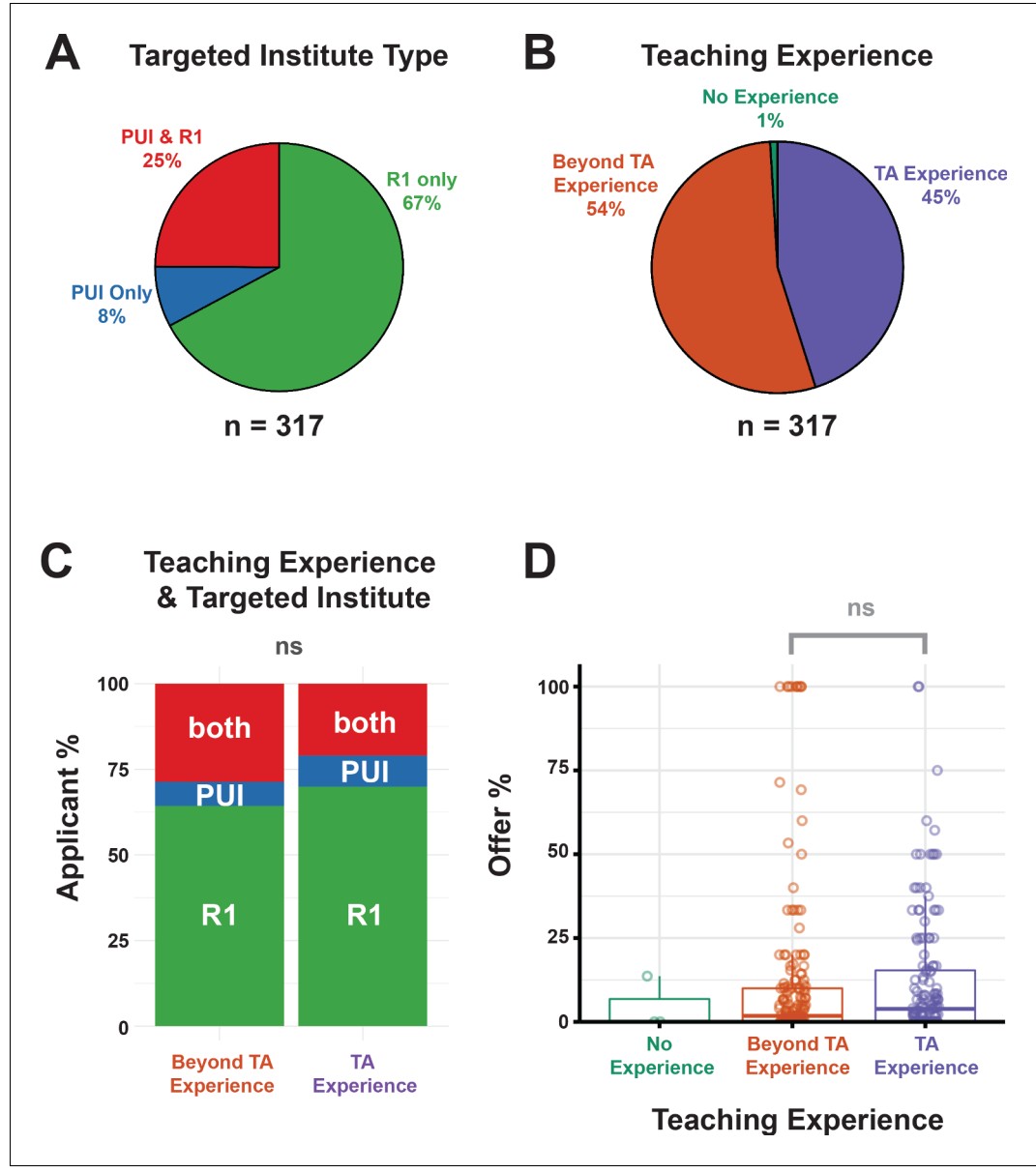

**Figure 6.** Summary of applicant teaching experience and impact on job search success. (**A**) Distribution of institution types targeted by survey applicants for faculty positions (PUI only in blue, R1 institutions only in green, or both in red, *Supplementary file 26*). (**B**) Distribution of teaching experience reported by applicants as having TA only experience (in purple), beyond TA experience (e.g. teaching certificate, undergraduate and/or graduate course instructorship, guest lectureship and college adjunct teaching, (in orange), or no teaching credentials (in green; *Supplementary files 27* and *28*). (**C**) Distribution of teaching experience (TA experience, right, vs. Beyond TA experience, left) for applicants who applied to R1 institutions only (in green), PU institutions only (blue), or both R1 and PUIs (in red), (*Supplementary file 27*). The degree of teaching experience did not change based on the target institution of the applicant (p=0.56 (ns), $\chi^2 = 0.41$; Chi-squared test). (**D**) Association between offer percentage and teaching experience is not significant (p=0.16; *Supplementary files 7*, *27* and *28*).

*Levels of teaching experience*

Discussions surrounding the academic job market often center on publications and/or funding, while teaching experience generally receives much less attention. However, the level of teaching experience expected from the applicants can vary, but mostly depends on the type of hiring institution.

We asked applicants whether they focused their applications to a specific type of institution (R1, PUI, or both; see *Box 1* for definitions), allowing us to examine teaching experience

## Box 1. Definition of specific terms used in this study.

**Early-career researcher (ECR):** For the purpose of this study, we define an ECR to be anyone engaged in research who is not recognized as an independent leader/investigator of a research group. This includes graduate and postdoctoral researchers; junior research assistants, research associates, and staff scientists.

**Principal Investigator (PI):** A scholar recognized as an independent leader of a research group. This includes full professors, group leaders, and tenure-track, non-tenure-track or tenured faculty.

**Faculty Job Applicant:** An early-career researcher with a PhD (a recent graduate, postdoctoral fellow or research scientist) who seeks to apply for a PI position (see above), usually at the assistant professorship level.

**STEM Fields:** STEM is an acronym for degrees in fields related to science, technology, engineering, and mathematics. STEM graduates work in a wide variety of fields including the life sciences, the physical sciences, different areas of engineering, mathematics, statistics, psychology, and computer science.

**Research Mentor:** A research advisor, usually the PI of a lab who mentors graduate and postdoctoral researchers during their academic training in his/her lab.

**Adjunct Lecturer:** A teacher or post-PhD scholar who teaches on a limited-term contract, often for one semester at a time. This individual is ineligible for tenure.

**Teaching Assistant (TA):** An individual who assists a course instructor with teaching-related duties in a lecture-based and/or laboratory-based undergraduate or graduate level course.

**Doctoral/Graduate and Postdoctoral Fellowships**: Funding mechanisms to support the training of a graduate or postdoctoral researcher: the proposal for this is written by the trainee and contains a mentoring/training plan and request for funding to support the trainee salary and/or part of their research expenses such as equipment, lab supplies and travel expenses typically for 1–3 years.

**Career Transition Awards:** Funding mechanisms facilitating senior trainees towards independent research careers: Includes core/substantial funds to fully support 1–3 years of postdoctoral salary and additional 2–5 years of independent faculty research and staff salaries as well as support for research expenses such as equipment, lab supplies and travel expenses. As a result, some portion of these funds can transition from the training institute to the hiring institute.

**R1 University:** There are 131 institutions in the United States that are classified as "R1: Doctoral Universities – very high research activity" in the Carnegie Classification of Institutions of Higher Education (2019 update), can be private or public.

**R2 University:** There are 135 institutions in the United States that are classified as "R2: Doctoral Universities – high research activity" in the Carnegie Classification of Institutions of Higher Education (2019 update), can be private or public.

**R3 University, PUI or Small Liberal Arts College (SLAC):** Primarily undergraduate institutions (PUI) are often smaller than large research universities, can be private or public, and offer varying levels of resources for students and faculty. Many faculty at PUIs run a research lab while maintaining significant teaching loads and heavy contact hours with students.

across R1 and/or PUI applicants. Most respondents applied to jobs at R1 institutions (*Figure 6A*), which may explain the focus on research-centric qualifications. It remains unclear what the emphasis on teaching experience is for search committees at R1 institutions, however the literature suggests that there seems to be a minimal focus (*Clement et al., 2019*). Additionally, there might be differences in departmental or institutional requirements that are unknown to outsiders. What is commonly accepted is that many applications to an R1 institution require a teaching philosophy statement.

Almost all respondents (99%) had teaching experience (*Figure 6B*): for roughly half this experience was limited to serving as a Teaching Assistant (TA; *Box 1*), with the rest reporting experience beyond a TA position, such as serving as an instructor of record (*Figure 6B*). The degree of teaching experience did not change based on the target institution of the applicant (*Figure 6C*), nor did the percentage of offers received significantly differ between groups based on teaching experience (*Figure 6D*).

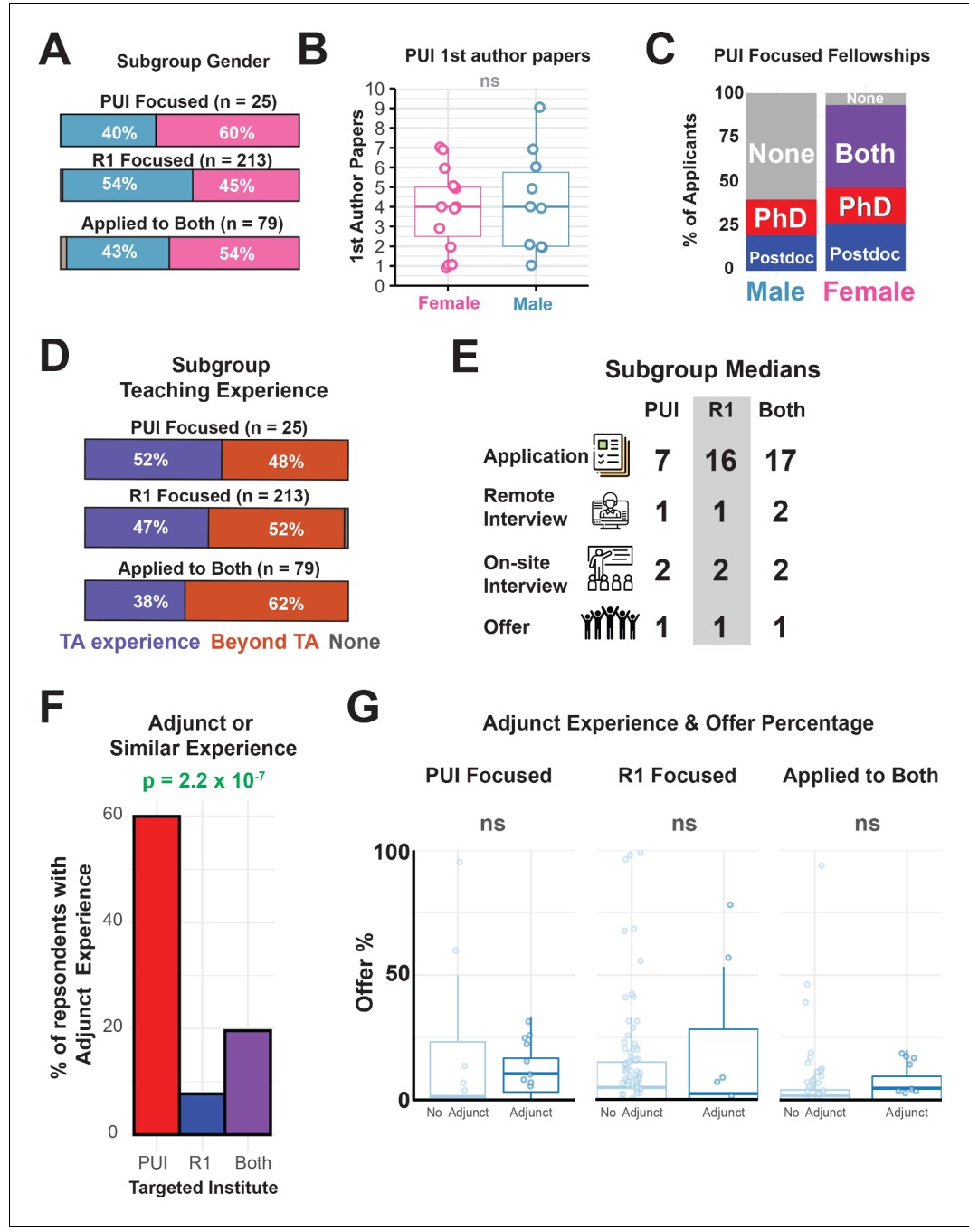

**Figure 7.** PUI focused applicants differ only in teaching experience from the rest of the application pool. (**A**) The gender distribution applicants who focused on applying to PUIs (*Supplementary file 26*). (**B**) The gender distribution and number of first-author publications of the applicant who focused on applying to PUIs (p=0.88). (**C**) Summary of the fellowship history by gender for PUI focused applicants (*Supplementary file 13*). (**D**) Distribution of teaching experience of PUI focused applicants (*Supplementary file 27*). (**E**) The median number of applications, off-site interviews, on-site interviews and offers for PUI focused applicants. (**F**) Percentage of survey respondents who identified having "adjunct teaching" experience (*Figure 1*) based on target institution (p=$5.0\times10^{-4}$; $\chi^2$ = 27.5, Chi-squared test). (**G**) The number of offers received segregated by "adjunct teaching" experience in either PUI focused applicants (p=0.55) or R1/both R1 and PUI focused applicants (p=0.98).

**Figure 8.** Perceptions of the job application process. Three word clouds summarizing qualitative responses from the job applicant survey respondents to the following questions: A) "What was helpful for your application? " (top; *Supplementary file 17*), (B) "What was an obstacle for your application? " (middle; *Supplementary file 18*), and C) "What is your general perception of the entire application process?" (bottom; *Supplementary file 31*). The size of the word (or short phrase) reflects its frequency in responses (bigger word corresponds to more frequency). Survey respondents were able to provide longer answers to these questions, as shown in *Supplementary files 17*, *18* and *31*. 'CNS-papers' refers to papers in Cell, Nature or Science; 'Pedigree' refers to the applicant's postdoc lab pedigree or postdoc university pedigree; 'Grant-Writing' refers to the applicant's grant writing experience with their PhD or postdoctoral mentor; 'Peer-reviewing' refers to the experience of performing peer-reviewing for journals; 'Interdisciplinary-research' refers to comments

*Figure 8 continued on next page*

## Research versus teaching-intensive institutions

To our knowledge, there is a lack of systematic evidence describing the process or expected qualifications of a PUI-focused (*Box 1*) job search (*Ramirez, 2016*). A subgroup of 25 "PUI Focused" applicants responded to our survey, and, despite this small number, we aimed to describe this important sub-group relative to "R1 Focused" applicants as well as applicants who applied to both types of institutes. The PUI subgroup included a majority of female applicants (60%, *Figure 7A*) while the R1 subgroup had a majority of male applicants (54%, *Figure 7A*). Within the PUI subgroup, no differences were seen in the number of first author publications across genders (*Figure 7B*), although women had a better fellowship history (*Figure 7C*). The median number of remote interviews, onsite interviews, and offers was also similar to that for the R1 subgroup, although the PUI subgroup submitted fewer applications (*Figure 7E*). Although both subgroups reported teaching experience (*Figure 7D*), the PUI subgroup was enriched in adjunct, visiting professor, instructor of record, community college, or contract-based teaching experiences (*Figure 7F*). Having adjunct experience did not significantly increase the median number of offers received for applicants focused on PUIs, R1s, or both types of institutions (*Figure 7G*).

## A time-consuming and opaque process with little feedback

We asked the applicants to comment on whether any aspect of their training or career was particularly helpful or harmful to their faculty applications (*Figure 8A–B*). We used word clouds (*Supplementary files 27* and *28*) to analyze recurrent themes in these open-ended questions. The applicants identified funding as most helpful for their applications, and no-funding as subsequently harmful; this perception agrees with the data presented above (*Figure 8A*, *Figure 5C*, *Figure 4—figure supplement 1*). Additionally, perceptions were also in line with the rest of the data, in that they were unable to largely agree on other measurable aspects of their career that were perceived as helpful. Qualitative aspects that were perceived as particularly helpful included networking and attending/presenting at conferences. Interestingly interdisciplinary-research, which is often highlighted as a strength and encouraged by institutions and funders, was perceived by

*Figure 8 continued*

stating that Interdisciplinary research was underappreciated; 'two-body problem' refers to the challenges that life-partners face when seeking employment in the same vicinity; 'No-Feedback' refers to lack of any feedback from the search committees on the status, quality or outcome of applications.

candidates as a challenge to overcome. Indeed, interdisciplinary candidates may pose an evaluation challenge for committees, given the differences in valuation of research metrics across fields, the extended training time required to master techniques and concepts in multiple fields, as well as valuation of interdisciplinary teams of specialists over interdisciplinary individuals (*Eddy, 2005*).

Notably, many applicants found the amount of time spent on applications and the subsequent lack of feedback from searches frustrating (*Figure 8B–C*). Most applicants never received any communication regarding their various submissions. For instance, an applicant who applied for 250 positions only received 30 rejections. Overall, our respondents submitted 7644 applications (*Figure 4A*) and did not hear anything back in 4365 cases (57% of applications), receiving 2920 formal rejection messages (38% of applications; *Supplementary file 19*). Application rejection messages (if received at all) most often do not include any sort of feedback. Additionally, a considerable amount of time is spent on writing each application and extensive tailoring is expected for competitive materials. Combining these insights, it is therefore unsurprising that almost all applicants, including applicants that received at least one offer (*Supplementary file 29*), found the process "time-consuming", a "burden on research", and "stressful" (*Figure 8B–C*).

44% of respondents had applied for faculty jobs for more than one cycle (*Supplementary file 30*). Though applicants who applied for more than one cycle had significantly lower offer percentages (p=3.45×10$^{-2}$; *Figure 5B*), many reported perceived benefits from significant feedback from their current PI through their previous application cycles. Though mentorship was not as often reported as specifically helpful (*Supplementary file 17*), the lack of mentorship was a commonly cited harmful obstacle (*Figure 8B*, *Supplementary file 18*). Lastly, multiple candidates felt that issues pertaining to family, the two-body problem (need for spousal/significant other hire), parental leave, or citizenship status significantly harmed their prospects.

## The view from the search committees

To learn more about the characteristics search committees valued in applicants, we performed an exploratory survey of members of such committees. This anonymous survey was distributed in a limited fashion, taking advantage of the professional networks of the authors. Fifteen faculty members responded, with nine having been involved in search committees for over ten years (*Figure 9A*). As with our survey of applicants, we focused on faculty members at R1 academic centers working in life sciences (14/15 of those polled) and engineering (1/15) within the United States (*Figure 9A*).

Two-thirds of respondents replied that the search committees they sat on typically received over 200 applicants per job posting, with one-third receiving 100–199 applications per cycle. Between 5 and 8 applicants were typically invited to interview on-site; one-third of respondents replied that off-site interviews (e.g., via phone or Skype) were not performed (*Figure 9B*). These statistics help demonstrate the challenges that hiring committees face; the sheer volume of applicants is overwhelming, as mentioned explicitly by several search committee respondents (*Supplementary file 32*).

We asked what factors search committee members found most important, what their perception of the market was, and how they felt it had changed since they first became involved in hiring. We also asked them to weigh specific application criteria in evaluating an application from 1 (not weighted at all) to 5 (heavily weighted; *Figure 9C*). Criteria such as transition awards were consistently ranked highly, matching applicant perception; however, committee members also placed substantial emphasis on the research proposal. Two-thirds viewed preprints favorably, although their strength may not yet be equivalent to published peer-reviewed work (*Figure 9C*). In follow-up questions, a number of respondents emphasized that the future potential of the candidate both as a colleague and a scientist was important.

Since this last point was not prominent in our survey of job applicants, we looked for discrepancies in the two sets of responses (*Figure 9D*). In general, search committees placed greater emphasis on the future potential and scientific character (research proposal, research impact, collegiality), while applicants focused on publication metrics and funding. However, despite the search committees placing less emphasis on

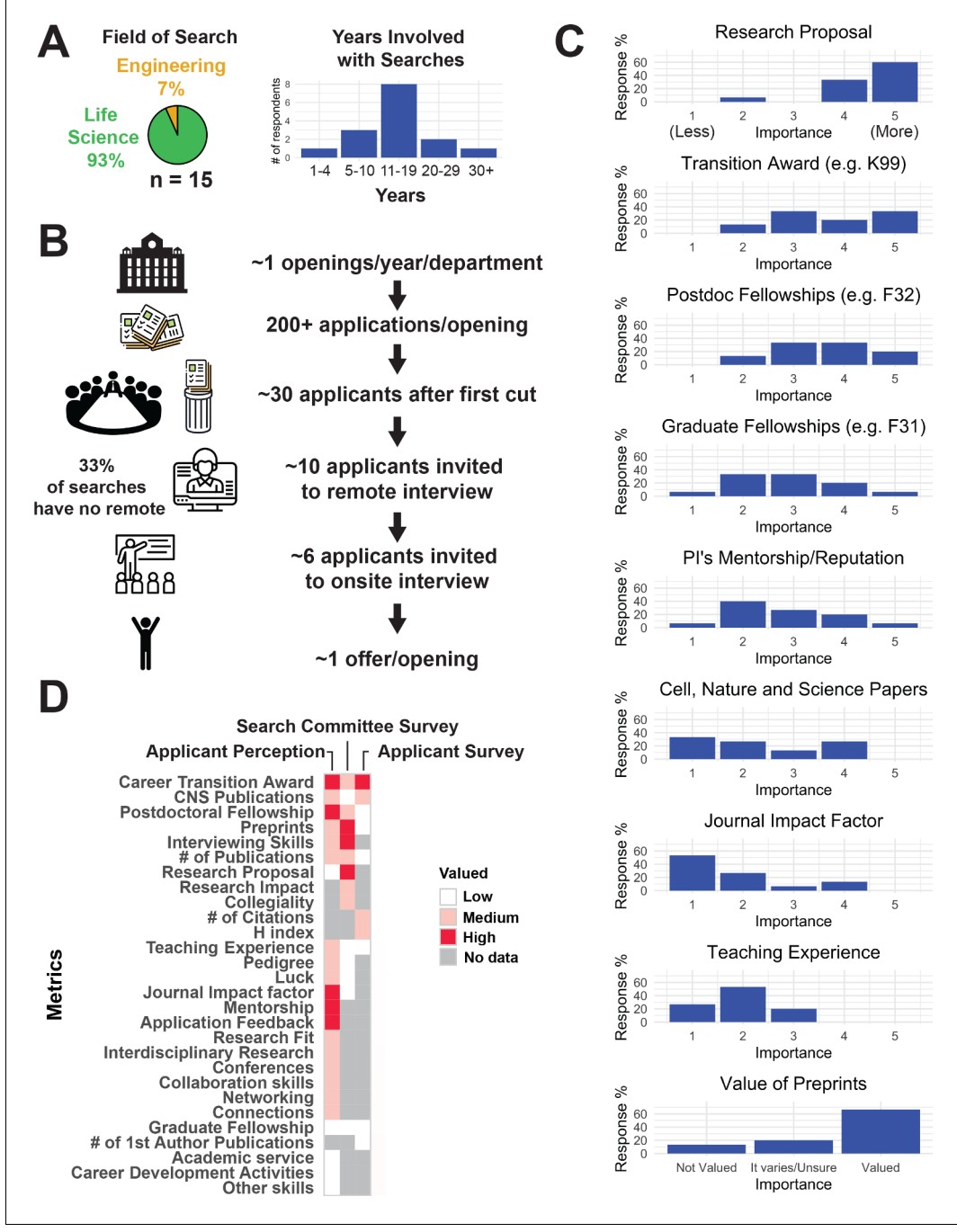

**Figure 9.** Summary of metrics valued by search committees. Search committee members were asked on how specific factors were weighted in the decision on which applicant to extend an offer to (***Supplementary files 33–38***). All search committee members surveyed were based at R1 universities (***Box 1***). (**A**) Distribution of the fields of study and years of experience for the search committee survey respondents. (**B**) The median number of faculty job openings, number of applicants per opening, applicants that make the first cut, applicants who are invited for phone/Skype interviews, and offers made. (**C**) The quantitative rating of search committee faculty on metrics: candidate/applicant research proposal, career transition awards, postdoctoral fellowships, graduate fellowships, PI/mentor reputation (lab pedigree), Cell/Nature/Science journal publications, Impact factor of other journal publications, Teaching experience and value of preprints based on a 5-level Likert scale where 1 = not at all and 5 = heavily. (**D**) Visual summary of the job applicant perception (from word cloud data) and the results of both surveys (statistical analyses of the applicant survey and criteria weighting from the search committee survey). A

*Figure 9 continued on next page*

*Figure 9 continued*

number of metrics mentioned in short answer responses were not measured/surveyed across all categories. These missing values are shown in gray.

papers in CNS, candidates with papers in these journals were more successful.

We also asked if there were additional factors that search committees wished applicants knew when applying (*Figure 10*). Several emphasized the quality of the research and papers was the most important factor for assessing prior achievement, but added that a compelling and coherent research proposal was also critical, and was sometimes underdeveloped in otherwise competitive candidates. The importance of departmental fit was also emphasized; interpersonal interactions with faculty members at the interview stage were also mentioned. This last sentiment is consistent with a recent Twitter poll which found that "overall attitude/vibe" was the single most important factor for selection at the interview stage (*Tye, 2019*). Intriguingly, while one faculty respondent noted that they rarely interview any applicant without a career transition award, such as a K99/R00 Pathway to Independence Award from the NIH (a situation they noted as problematic), another lamented that applicants worried too much about metrics/benchmarks anecdotally perceived to be important, such as receiving these awards. Finally, a majority of respondents noted that it was easy to identify good candidates from their submitted application (11/15), that there were too many good applicants (10/15), and that candidates often underperformed at the interview stage (10/15) (*Figure 10*, *Figure 10—figure supplement 1*, *Supplementary File 35*).

## Discussion

### Challenges in the academic job market

Currently, there is little systematic evidence for what makes a competitive faculty candidate. As with any opaque, high-pressure environment, an absence of clear guidelines and expectations coupled with anecdotal advice can lead individuals to focus on tangible goals and metrics that they feel will help them stand out in the system. Our findings were consistent with several commonly held notions: the number of applications submitted, career transition awards (e.g. a K99/R00 award), and total citation counts were significantly associated with obtaining offers in our Wilcoxon test and when jointly considering all variables in a logistic regression analysis. Joint academic/industry job searches were negatively associated with obtaining academic offers in both analyses, while the number of years an applicant was on the job market was negatively associated in our Wilcoxon analysis. Papers in CNS were only significantly associated with offers in the Wilcoxon analysis, while postdoc fellowships were only significant in the logistic regression.

Metrics such as career transition awards and postdoctoral fellowships can be broadly categorized as funding metrics and the positive association between these metrics and offer outcomes likely reflects the hiring institute being confident that the candidate will be competitive for future funding for their research program. Indeed, career transition awards essentially provide additional start up funds, while postdoc fellowships provide a track record of funding. Although postdoc fellowships were not significant in our Wilcoxon analyses, this metric was significant in our life science-specific Wilcoxon subgroup analysis (*Figure 5—figure supplement 1*.) as well as our logistic regression on the whole dataset (*Figure 5D*). The search committee respondents confirmed the benefit of career transition funding as major strengths for an application.

Association between offers and the number of applications, non-academic job searches, and years on the academic job market requires cautious interpretation. Given that receiving any single faculty offer is a low-probability event, there is value in submitting enough applications to increase the odds of receiving an offer. However, there is likely a balance in ensuring the quality of each application, which requires time and effort to individually tailor to each position. Searching for non-academic jobs might detract from the time available to tailor applications, although the negative association may also reflect other factors such as the typically swifter non-academic hiring timeline, which could cause applicants to remove themselves from a search prior to its conclusion. Likewise, the negative association between repeated years on the job market and offers might reflect fundamental problems with the quality of an application, or more complex factors such as geographical constraints. As we did not collect data that would allow us to determine the quality of application,

**Figure 10.** Search committee perception of the faculty job application process. Two word clouds representing responses from members of search committees in response to the following questions: A) "What information do you wish more candidates knew when they submit their application?", and B) "Have you noticed any changes in the search process since the first search you were involved in?" The size of the word/phrase reflects its frequency in responses, with larger phrases corresponding to more frequent responses. Search committee faculty members were able to provide long answers to both questions (*Supplementary files 38* and *39*).

The online version of this article includes the following figure supplement(s) for figure 10:

**Figure supplement 1.** Overview of search committee impressions of the candidates.

or the fit of an application to a particular opening, we cannot evaluate these metrics beyond the broad associations found in our dataset. Additionally, other unmeasured factors (e.g. applicant pedigree) are likely important considerations, consistent with recent data implicating institutional prestige and non-meritocratic factors in faculty hiring (*Clauset et al., 2015*). This should be a major consideration for future studies of the academic job market.

When examining publication-related metrics, we found that total citation counts were significantly associated with receiving a job offer in both the Wilcoxon and logistic regression analyses. There was also a significant positive association between being first author on a CNS paper and receiving a job offer in the Wilcoxon

analysis, but not in our logistic regression models. Examination of our data also revealed a gender gap in publication metrics, with males reporting more CNS papers and more papers overall, indicating that opportunities for publication are not equally available (*Arvanitis and Cho, 2018*; *Gumpertz et al., 2017*). Second, the results of our automated variable selection procedure suggest that being an author in any position on a paper in CNS is an advantage overall (though the result is not significant); however, within the life sciences, being the first author is more of an advantage (again, not significant). Finally, papers in CNS and other journals with high impact factors have been regarded as a major benchmark for trainees in the life sciences (*van Dijk et al., 2014*), and qualitative comments from our applicant survey conveyed a perception that the absence of a CNS paper is deemed detrimental to offer prospects. Collectively, our data suggest that while being first author on a CNS paper increases the chances of receiving an offer (particularly in life sciences), papers in CNS were neither necessary nor sufficient for securing an offer, as the majority of our respondents received offers without having a paper in CNS.

Consistently, being the author of a CNS paper was not deemed highly important by the search committee members we surveyed. These data may reflect a discordance of priorities for individual faculty members compared to their peers and the system at-large, as recently reported (*Niles et al., 2019*). This could lead to an unspoken expectation that faculty (especially pre-tenure faculty) see themselves as passive participants in the current academic system, instead of active participants with the authority to realign priorities through search committees (*Niles et al., 2019*). Future studies with higher numbers of faculty respondents should endeavor to further explore this phenomenon.

Despite challenges in the job market (*Larson et al., 2014*; *Andalib et al., 2018*; *Kahn and Ginther, 2017*), our survey revealed positive outcomes that suggest progress in select areas. Nearly half of the job applicants we surveyed reported posting at least one preprint. Several of the search committee members we surveyed confirmed that while published papers carry the most weight, preprints are generally viewed favorably. Further, despite the fact that women face numerous challenges in academia, including underrepresentation at the faculty level in most STEM departments (*Arvanitis and Cho, 2018*; *Gumpertz et al., 2017*; *Ceci and*

*Williams, 2015*; *Leaper and Starr, 2019*), and trail men in publication-related metrics (*Figure 3B*), our data suggest very few differences in outcomes in the May 2018–May 2019 job cycle. Both genders received similar numbers of interviews and offers, and gender-based differences in publication-related metrics persisted even when considering only the 185 individuals with offers, suggesting that committees are becoming increasingly aware of gender bias in publication-related metrics and are taking them into account when evaluating applicants (*Supplementary file 40*).

Overall, the respondents were generally highly qualified according to the metrics we measured, and yet they reported high stress and frustration with their experiences of the faculty job search. In a large number of cases, applicants were not notified of a receipt of their application, nor were they updated on its status, given a final notice of rejection, or informed that the search may have failed. This uncertainty further complicates an already stressful process that can be mitigated by improving practices for a more streamlined application process. Applicants perceived poor mentorship as a major obstacle to their applications. Further, we found that most metrics were differentially valued by candidates and committees. Collectively, these differences in expectations between applicants and hiring institutions, coupled with the opaque requirements for obtaining a faculty position, likely drive the high stress reported by both candidates and committee members alike.

### Limitations of this study and measuring outcomes in the academic job market

There are several limitations of this study imposed by both the original survey design and general concerns, such as the anonymity of respondents, and the measurability of various contributing factors. For future data collection we suggest keeping surveys focused on region-specific job markets. Our pool of applicants was largely those seeking a position in North America. We believe these results can be aggregated, but the survey questions may not all be applicable to other large markets (e.g. Europe, China, India). We did not receive a sizable response from applicants looking outside of North America and in fields outside of life sciences to make useful comparisons. A similar survey circulated in each market individually with a similar number of responses would have broader impact.

We purposely did not ask for race, ethnicity, or citizenship demographics, PhD or postdoc institution, and region or institution where offers were received. We believe the addition of these metrics could potentially jeopardize the anonymity of respondents. Despite this, these factors could be significant contributors to the receipt of an academic job offer. Racial inequalities in all STEM fields at all levels exist and need to be addressed (*Whittaker et al., 2015*), specifically with how they intersect with gender (*Gumpertz et al., 2017*). As indicated in our open question responses (*Figure 8B*), international postdocs may be specifically challenged in obtaining faculty job offers in the United States and Europe due to immigration policies as well as how mobility is interpreted by the job market (*Cantwell, 2011*). The reputation of a training institution is questionably measurable, but is also often listed in anecdotal advice as important. Recently it was reported that a majority of new faculty are hired from a minority of institutions providing postdoc training (*Clauset et al., 2015*; *Miuccio et al., 2017*). It is possible that adding institutional reputation to the other traditional metrics we measured could provide a more complete picture of the current path to a faculty position.

While we measured some of the attributes widely perceived as important in faculty hiring (e.g. funding track record), others are less easily quantified (e.g. the research proposal, lab pedigree, or letters of recommendation that comments from our search committee survey revealed to be important) and data collection on these items would be highly recommended in future surveys. Addressing the quality of application materials is highly context-specific (given the field, search committee, and institutional needs) and can improve (*Grinstein and Treister, 2018*). Other aspects which are not directly measurable and are often cited as important for applicants in the academic job market are "fit" and "networking" (*Wright and Vanderford, 2017*). Respondents agreed that networking, conferences, collaborations, and connections were helpful in their job search (*Figure 8A*). Conference organizers are also starting to offer badges that those searching for faculty jobs can wear at events; exploring the relationship between networking metrics (such as number of conferences and networking events attended) and success on the job market could be a topic for future research. Departmental or institutional "fit" is largely determined by the search committee on an individual basis, and it is likely that we will never be able to measure fit adequately (*Saxbe, 2019*).

All questions in our survey were optional. We chose this survey design in order to make the survey easier for respondents to complete; however, missing answers represent a source of potential bias as unanswered questions may represent answers that could be negatively perceived and/or zero in value. For example, some individuals may not have felt comfortable indicating they had zero offers, which could lead to the offer percentages we report being inflated. Such bias could also affect the imputations in our logistic regression, and for these reasons we have attempted to provide multiple transparent and qualified analyses of the data. Future surveys may benefit from all questions requiring a response. It is also possible that participation in the survey from the outset suffers from survivorship bias, in that those applicants that had a positive experience are more likely to reflect upon it and complete a survey on the process. Our survey was also likely completed by a highly-engaged group of aspiring future faculty. The Future PI Slack group itself is a space for postdoctoral researchers most interested in obtaining a faculty career to engage with and learn from one another. Thus, the survey data likely reflects a highly motivated and accomplished group and not the full pool of applicants to faculty positions each year. Wider dissemination of future surveys will hopefully be aided by the publication of these results and increased awareness of the survey among trainees in various research communities.

Finally, the data from our survey of job applicants focused on candidates and not the search committees. It is unclear how many individual searches are represented in our dataset. It is likely that as many as ~200–500 committees were represented in our aggregated job applicant data, and different committees may adopt distinct assessment criteria. Our limited search committee survey responses show that the committees represented by our sample favor a holistic assessment of candidates and that decision by universal criteria (especially based solely on career transition awards or papers in CNS) is likely not unilateral, especially across disciplines. Future studies would benefit from surveying a larger pool of search committees to see what major trends and practices dominate, whether the majority of searches adopt a comprehensive evaluation approach, or if there is heterogeneity among committees in how tenure-track hiring assessments are conducted.

## Conclusion

The search process for faculty jobs lacks transparency and data regarding what makes a successful applicant. Here, we began to address this deficiency through a survey targeted at the applicants themselves, and including their perceptions of the application process. Of over 300 responses by job applicants, we did not receive a single positive comment about the process, despite the fact that 58% of our participants received at least one job offer. Our data suggest that baseline thresholds exist for those more likely to receive a faculty job offer, but that many different paths can lead to a job offer. This variety of paths likely reflects both the preparation done by applicants and the different evaluation criteria used by individual search committees. For these reasons, we urge applicants not to conclude that lower than average metrics in any one area are automatically disqualifying. Indeed, we believe that increasing the transparency of the application process through systematic data collection will allow a more detailed study of the many paths to obtaining a faculty offer.

Our data also show the mental strain on applicants during the hiring process. We propose a number of potential solutions with the understanding that hiring faculty is a complex process involving multiple stakeholders. We believe the application process could be improved by simplifying the process, including standardizing application materials (e.g. requirements for research statements are similar for R1 institutions) and requesting references only after candidates are shortlisted, so that the burden of application preparation time can be reduced. Constructive feedback from mentors is vital for success during the application and interview preparation stages. Additionally, if possible, communication from search committees about unsuccessful applications would be helpful. We understand that these points may increase the workload of mentors and search committees but, if put into place, could alleviate some of the stress related to the academic job application process. In addition, applicants need to work to be sure their materials are strong and well-researched as the quality of these materials and demonstrating fit for a job posting are important to faculty on search committees (*Clement et al., 2019*). Further work into the challenges search committees face is needed to improve their experience of the application process.

It is our hope that this and future work will not only allow all stakeholders to make informed decisions, but will also enable critical examination, discussion, and reassessment of the implicit and explicit values and biases being used to select the next generation of academic faculty. Such discussions are crucial in building an academic environment that values and supports all of its members.

## Materials and methods

### Survey materials

We designed a survey (the "applicant survey") to collect demographics and metrics that were commonly discussed on Future PI Slack during the 2018–2019 academic job search cycle. The survey was designed to take less than 5 min in order to maximize response rates, and respondents were not required to answer all questions.

After collecting and performing initial analyses of this survey, we designed an additional survey for search committees (the "search committee survey"). The text of both surveys used in this work is included in the *Supplementary files 41* and *42*. A Google form was used to conduct both surveys.

The applicant survey was distributed on various social media platforms including the Future PI Slack group, Twitter, and Facebook, and by several postdoctoral association mailing lists including in North America, Europe and Asia. The survey was open for approximately six weeks to collect responses.

The search committee survey was distributed to specific network contacts of the various authors. Though this distribution was more targeted, a Google form link was still used to maintain anonymity. The search committee survey was open for approximately three weeks to collect responses. In both cases, respondents to the surveys were asked to self-report, and the information collected was not independently verified. The surveys can be found in *Supplementary files 41* and *42*.

### Data analysis

Prior to analysis, we manually filtered out five responses in which answers were not interpretable or did not appear to answer the correct questions. Microsoft Excel and RStudio were used to graph the results of both surveys shown in *Figures 1–6* and *8*. Specifically, data was filtered and subdivided using the 'tidyverse' collection of R packages, and figure plots were generated using the 'ggplot2' package. Whenever statistical analyses were used, the exact tests, p-values and $\chi^2$ values are reported in the appropriate figure or figure legend or caption, results section and *Supplementary file 7*, and represent the implementations in the basic R 'stats' package.

A p-value of less than 0.05 was considered significant. Where a number of demographics are combined in the reporting throughout this study, any analysis groups with less than five respondents were combined with other similar values instead of the raw *n* value in an effort to protect the anonymity of participants. Briefly, statistical methods are as follows: in general, the two-tailed Wilcoxon rank sum test (with Holm correction when applicable) or Chi-squared test was used to report p-values (see *Supplementary file 7* for detailed breakdown).

The qualitative survey comments were categorized by theme (keywords/context) describing each comment and the frequency of comments pertaining to a particular theme and tabulated (*Supplementary files 17*, *18*, *38* and *39*). Word clouds were generated using the WordItOut platform (*WordItOut, 2020*; *Figures 7* and *9*). The visual summary heatmap of the job applicant perception and the survey results along with the search committee survey results (*Figure 9D*) was created by counting the frequency of comments for each metric (i.e. publications, fellowships, preprints) from the respondents to the qualitative (long answer) questions (*Supplementary files 17*, *18*, *38* and *39*). The job applicant survey quantitative results were also used to rank metrics based on significance (as determined by Wilcoxon analysis or logistic regression analysis (*Supplementary file 7*)) and were also incorporated into the heatmap (*Figure 9D*). A number of metrics were not measured/surveyed as part of our study. These missing values are shown in gray.

Logistic regression analysis was performed in R using the 'glm' function with the 'family' parameter set to 'binomial'. All variables collected in the survey were included as independent variables, except those that were considered to be outcomes (numbers of remote interviews, onsite interviews and offers). The outcome variable was a binary 'Offer' or 'No offer' variable. All continuous variables were z-score normalized to ensure that they were centered and scaled consistently. To reduce collinearity between variables, a forward stepwise variable selection approach was adopted by starting with the variable that was most accurate in predicting

offer status when included in a logistic regression model and then iteratively adding a variable to the model to maximize accuracy at every step. Furthermore, at every step, a variable would only be added if it was not correlated (Spearman correlation coefficient $\leq$0.5) with a variable already included in the model from a previous step. The model with the most accurate variable-combination was used to report coefficients. When multiple independent variables were considered together, missing values accounted for nearly two-thirds of the data, and were therefore imputed by fitting a bagged tree model for each variable (as a function of all the others; 63). Both variations of the analysis (missing data excluded and missing data imputed) were reported. In addition, this entire logistic regression analysis was repeated on a subset, solely comprising of applicants from the life sciences.

In order to visualize the potential paths to an offer, a decision tree was learned automatically from the data using the C5.0 algorithm (*Kuhn and Johnson, 2013*). All possible combinations of the following parameter settings were evaluated: (*Cyranoski et al., 2011*) either the tree-based variant or the rule-based variant of the algorithm was run, (*Ghaffarzadegan et al., 2015*) winnowing of irrelevant variables was set to 'TRUE' or 'FALSE', and (*Schillebeeckx et al., 2013*) the number of boosting 'trials' was set to 1, 4, 16, 32 or 64. The parameter combination with the best accuracy in predicting offer status in a 10-fold cross-validation experiment (as implemented in the 'caret' package in R) was chosen (*Kuhn, 2008*). Since decision trees naturally handle missing values and differences in scales, no additional imputation or data normalization was performed before training and testing. The most accurate tree was found to be the one that used the rule-based variant, had no winnowing and no boosting (trials = 1) and was plotted using the 'plot' function in the 'partykit' R package (*Hothorn and Zeileis, 2015*) and then manually grouped in Illustrator.

## Acknowledgements
The authors thank Carol Greider, Feilim Mac Gabhann, Cori Bargmann, Mark Kunitomi, Needhi Bhalla, Lucia Peixoto, Sarah Stone and Dario Taraborelli for their valuable comments on an earlier version of this manuscript. The authors are members of the Future PI Slack community and would like to thank the entire Future PI Slack community and those who support them in this work.

**Jason D Fernandes** is in the Department of Biomolecular Engineering, University of California, Santa Cruz, United States and is a member of the eLife Community Ambassadors programme
https://orcid.org/0000-0002-8625-1796

**Sarvenaz Sarabipour** is in the Institute for Computational Medicine and the Department of Biomedical Engineering, Johns Hopkins University, Baltimore, United States and is a member of the eLife Early-Career Advisory Group
https://orcid.org/0000-0001-5097-5509

**Christopher T Smith** is in the Office of Postdoctoral Affairs, North Carolina State University Graduate School, Raleigh, United States
https://orcid.org/0000-0002-8212-7886

**Natalie M Niemi** is in the Morgridge Institute for Research, Madison, United States and in the Department of Biochemistry, University of Wisconsin-Madison, Madison, United States
https://orcid.org/0000-0002-5174-4005

**Nafisa M Jadavji** is in the Department of Biomedical Sciences Midwestern University, Glendale, United States and is a member of the eLife Community Ambassadors programme
https://orcid.org/0000-0002-3557-7307

**Ariangela J Kozik** is in the Division of Pulmonary and Critical Care Medicine, Department of Internal Medicine, University of Michigan, Ann Arbor, United States
https://orcid.org/0000-0003-2322-4085

**Alex S Holehouse** is in the Department of Biochemistry and Molecular Biophysics, Washington University School of Medicine, St. Louis, United States
https://orcid.org/0000-0002-4155-5729

**Vikas Pejaver** is in the Department of Biomedical Informatics and Medical Education and the eScience Institute, University of Washington, Seattle, United States
https://orcid.org/0000-0002-1943-0284

**Orsolya Symmons** was at the Department of Bioengineering, University of Pennsylvania, Philadelphia, United States. Current address: Max Planck Institute for Biology of Ageing, Cologne, Germany
https://orcid.org/0000-0003-2435-4236

**Alexandre W Bisson Filho** is in the Department of Biology and the Rosenstiel Basic Medical Science Research Center, Brandeis University, Waltham, United States
https://orcid.org/0000-0002-5940-7230

**Amanda Haage** is in the Department of Biomedical Sciences, University of North Dakota, Grand Forks, United States
amanda.haage@und.edu
https://orcid.org/0000-0001-6305-440X

*Author contributions:* Jason D Fernandes, Conceptualization, Resources, Data curation, Software, Formal analysis, Validation, Investigation, Visualization,

Methodology, Writing - original draft, Project administration, Writing - review and editing; Sarvenaz Sarabipour, Conceptualization, Data curation, Formal analysis, Validation, Investigation, Visualization, Methodology, Writing - original draft, Writing - review and editing; Christopher T Smith, Validation, Investigation, Methodology, Writing - original draft, Writing - review and editing; Natalie M Niemi, Nafisa M Jadavji, Conceptualization, Investigation, Methodology, Writing - original draft, Writing - review and editing; Ariangela J Kozik, Validation, Investigation, Methodology, Writing - original draft, Project administration, Writing - review and editing; Alex S Holehouse, Conceptualization, Resources, Data curation, Investigation, Methodology, Project administration; Vikas Pejaver, Conceptualization, Data curation, Software, Formal analysis, Validation, Visualization, Writing - review and editing; Orsolya Symmons, Conceptualization, Data curation, Writing - review and editing; Alexandre W Bisson Filho, Data curation, Formal analysis, Validation, Investigation, Visualization, Methodology; Amanda Haage, Conceptualization, Resources, Data curation, Formal analysis, Supervision, Funding acquisition, Validation, Investigation, Visualization, Methodology, Writing - original draft, Project administration, Writing - review and editing

**Competing interests:** The authors declare that no competing interests exist.

**Ethics:** Human subjects: This survey was created by researchers listed as authors on this publication, affiliated with universities in the United States in an effort to promote increased transparency on challenges early career researchers face during the academic job search process. The authors respect the confidentiality and anonymity of all respondents. No identifiable private information has been collected by the surveys presented in this publication. Participation in both surveys has been voluntary and the respondents could choose to stop responding to the surveys at any time. Both 'Job Applicant' and 'Search Committee' survey has been verified by the University of North Dakota Institutional Review Board (IRB) as Exempt according to 45CFR46.101(b)(2): Anonymous Surveys No Risk on 08/29/2019. IRB project number: IRB-201908-045. Please contact Dr. Amanda Haage (amanda.haage@und.edu) for further inquiries.

**Funding**

| Funder | Grant reference number | Author |
| --- | --- | --- |
| University of North Dakota | Start-up funds | Amanda Haage |
| National Institute of General Medical Sciences | F32GM125388 | Jason D Fernandes |
| National Heart, Lung, and Blood Institute | T32HL007749 | Ariangela J Kozik |
| Midwestern University | Start-up funds | Nafisa M Jadavji |
| Washington Research Foundation | Fund for Innovation in Data-Intensive Discovery | Vikas Pejaver |
| University of Washington | Moore-Sloan Data Science Environments Project | Vikas Pejaver |
| Washington University in St. Louis | Start-up funds | Alex S Holehouse |

The funders had no role in study design, data collection and interpretation, or the decision to submit the work for publication.

**Decision letter and Author response**
Decision letter https://doi.org/10.7554/eLife.54097.sa1
Author response https://doi.org/10.7554/eLife.54097.sa2

## Additional files
### Supplementary files
• Supplementary file 1. Common online resources for finding academic jobs. Resources for finding academic jobs, often mentioned by our applicant survey respondents and cited by others as helpful for locating academic job announcements across different fields.

• Supplementary file 2. Applicants by field of research and gender. Overview of job application survey respondents' (total and by gender) field of study. Fields which had fewer than three respondents in our job applicant survey were aggregated as "Other Fields" in the table. All percentages are calculated out of the total number of respondents.

• Supplementary file 3. Applicant demographics: country of research origin (applicant location). Overview of candidates' country of research origin. Regions which had fewer than five respondents in our job applicant survey were aggregated as "Other countries" in the table. All percentages are calculated out of the total number of respondents to this particular survey question (297) not the total number of overall survey respondents (n = 317).

• Supplementary file 4. Country to which faculty application was made (job location). Overview of the countries to which the faculty candidates applied to, for faculty positions. Note: most candidates applied to more than one country. Regions which had fewer than five respondents in our job applicant survey were aggregated as "Other countries and regions" in the table. All percentages are calculated out of the total number of respondents to this particular survey question (n = 317).

• Supplementary file 5. Current research/academic position for all applicants. Overview of current academic position of our job applicant survey respondents. All percentages are calculated out of the total number of respondents to this particular survey question (n = 317).

• Supplementary file 6. Postdoctoral training times for all applicants. Overview of time spent in postdoctoral training by our job applicant survey respondents.

• Supplementary file 7. Summary of the statistical analysis in this paper. Summary of statistical analysis. In this table and relevant figures, **"ns"** stands for not significant.

• Supplementary file 8. Applicant demographics: applicants with first or multiple postdoctoral position. Overview of number of postdoctoral positions that the candidates held at the time of their faculty job application. All percentages are calculated out of the total number of respondents to this particular survey question.

• Supplementary file 9. Scholarly metrics for all applicants. Overview of the job applicant publication metrics (average citation number, average h-index, average number of peer-reviewed papers, average number of preprints, average number of peer-reviewed first-author papers, number of Cell/Nature/Science journal publications or "CNS" papers of any type meaning first author, co-author or corresponding author) of our survey respondents by gender breakdown.

• Supplementary file 10. Scholarly metrics for applicants in the life/biomedical sciences. Overview of the job applicant publication metrics (average citation number, average h-index, average number of peer-reviewed papers, average number of preprints, average number of peer-reviewed first-author papers, number of Cell/Nature/Science journal publications or "CNS" papers of any type meaning first author, co-author or corresponding author) of our survey respondents in life/biomedical sciences (respondents who indicated their field of research as Chemistry, Biology, Bioengineering or Biomedical or Life Sciences) by gender breakdown.

• Supplementary file 11. Responses on Cell/Nature/Science or "CNS" journal publications for all applicants. Overview of the number of Cell/Nature/Science ("CNS") journal publications of our job applicant survey respondents by gender breakdown. Percentages are calculated out of the total number of respondents to this particular survey question.

• Supplementary file 12. Responses on Cell/Nature/Science or "CNS" journal publications from applicants in the life/biomedical sciences. Overview of the number of Cell/Nature/Science ("CNS") journal publications of our job applicant survey respondents in life/biomedical sciences (respondents who indicated their field of research as Chemistry, Biology, Bioengineering or Biomedical or Life Sciences) by gender breakdown.

Percentages are calculated out of the total number of respondents to this particular survey question.

• Supplementary file 13. Fellowships and funding. Overview of the types of funding held by our job applicant survey respondents. Percentages are calculated out of the total number of respondents to this particular survey question. All percentages are calculated out of the total number of respondents to this particular survey question. Our survey questions did not distinguish between the types (e.g. government funded vs privately funded, full vs partial salary support) or number of fellowships applied to; many of these factors are likely critical in better understanding gender differences in fellowship support.

• Supplementary file 14. Responses about preprints. Overview of candidates who had unpublished preprints at the time of their job application. Percentages are calculated out of the total number of respondents to this particular survey question.

• Supplementary file 15. Twitter poll: number of offers current faculty received. Overview of the responses to a twitter poll with the question: "Faculty, when you accepted your first position, how many offers did you have to choose from?"

• Supplementary file 16. Applicants who also applied to non-faculty jobs. Overview of candidates who also applied for non-faculty jobs (e.g. Industry positions, government jobs, etc.). Percentages are calculated out of the total number of respondents to this particular survey question (n = 315 applicants).

• Supplementary file 17. Themes from job applicant survey written responses to helped your application. Candidate responses to "Was any aspect of your career particularly helpful when applying (preprints, grants etc.)?" Survey participants were able to provide long answers to this comment question. A word cloud referring to this table of comments is provided in *Figure 8A*.

• Supplementary file 18. Themes from written responses to question about obstacles. Candidate responses to "Was any aspect of your career particularly an obstacle when applying?" Survey participants were able to provide long answers to this comment question. A word cloud referring to this table of comments is provided in *Figure 8B*.

• Supplementary file 19. Application statistics. Overview of application statistics: total number of applications made, offsite (remote via phone or online via Skype) interviews, onsite interviews, offers made, approximate number of rejections and total number of no feedbacks received from faculty job committees to our survey respondents.

• Supplementary file 20. Career transition awards. Overview of the types of transition/independent type funding held by our faculty candidate (applicant survey) respondents. Percentages are calculated out of the total number of respondents to this particular survey question. Being a 'Co-PI' of a grant as a

postdoctoral researcher or research scientist means co-writing a grant with a PI (an independent investigator). The co-writer may or may not be explicitly mentioned on the grant as a Co-PI.

• Supplementary file 21. Responses on patenting. Overview of Candidates who had approved or pending patents from their research at the time of their job application. Percentages are calculated out of the total number of respondents to this particular survey question.

• Supplementary file 22. Use of resources that offered information about the application process. Overview of candidates who were familiar with the Future PI Slack resource and other resources during their application process. Responses to "Did you find the Future PI google sheet/Slack helpful? Yes/No" Survey participants were able to provide a long answer to this comment question (Future PI Slack or FPI Slack is a Slack group comprised of postdoctoral researchers aspiring to apply for faculty/Principal Investigator positions).

• Supplementary file 23. Responses to "Why did you find the Future PI Google Sheet helpful?". Overview of candidates who were familiar with the Future PI Slack resource and other resources during their application process. Responses to "Why did you find the Future PI google sheet/Slack helpful?" Survey participants were able to provide a long answer to this comment question. Note: Future PI Slack is a Slack group of postdoctoral researchers who aspire to apply for faculty positions.

• Supplementary file 24. Logistic regression with stepwise variable selection analysis on the survey data. Regression analysis with stepwise variable selection was performed on the job applicant survey data. All variables collected except for the number of remote and on-site interviews were included as potential predictors of receiving (*Cyranoski et al., 2011*) or not receiving (0) a job offer. Positive coefficients indicate positive associations and negative coefficients indicate negative associations with receiving an offer. Coefficients that are zero indicate no association. Bold values indicate that the associations were found to be significant at a threshold of 0.05. Summary of results testing criteria with offer outcomes either through Wilcoxon analyses or logistic regression. When applicants with missing values were excluded, application number ($\beta$=0.5345, p=1.53×10$^{-3}$), having a postdoctoral fellowship ($\beta$=0.4013, p=6.23×10$^{-3}$), and number of citations ($\beta$=0.4178, p=2.01×10$^{-2}$) positively associated with offer status in a significant manner, while searching for other jobs ($\beta$=−0.3902, p=1.04×10$^{-2}$) negatively associated with offer status in a significant manner. When missing values were imputed, significant positive coefficients were observed for application number ($\beta$=0.5171, p=8.55×10$^{-4}$), funding ($\beta$=0.3156, p=1.72×10$^{-2}$), having a postdoctoral fellowship ($\beta$=0.2583, p=3.75×10$^{-2}$) and citations ($\beta$=0.4363, p=1.34×10$^{-2}$). Moreover, the search for non-academic jobs ($\beta$=−0.2944, p=1.98×10$^{-2}$) and the number of

years on the job market ($\beta$=−0.2286, p=7.74×10$^{-2}$) were significantly negatively associated with offer status.

• Supplementary file 25. Logistic regression with stepwise variable selection analysis on survey data of applicants from only the life sciences applicants. Regression analysis with stepwise variable selection was performed on the subset of the job applicant survey data corresponding to applicants from the life sciences. All variables collected except for the number of remote and on-site interviews were included as potential predictors of receiving (*Cyranoski et al., 2011*) or not receiving (0) a job offer. Positive coefficients indicate positive associations and negative coefficients indicate negative associations with receiving an offer. Coefficients that are zero indicate no association. Bold values indicate that the associations were found to be significant at a threshold of 0.05. Summary of results testing criteria with offer outcomes either through Wilcoxon analyses or logistic regression. When applicants with missing values were excluded, application number ($\beta$=0.5827, p=1.07×10$^{-3}$) and having a postdoctoral fellowship ($\beta$=0.5738, p=1.74×10$^{-3}$) positively associated with offer status in a significant manner, while searching for other jobs ($\beta$=−0.3975, p=3.16×10$^{-2}$) negatively associated with offer status in a significant manner. When missing values were imputed, significant positive coefficients were observed for application number ($\beta$=0.5445, p=4.54×10$^{-4}$), funding ($\beta$=0.3687, p=1.27×10$^{-2}$), having a postdoctoral fellowship ($\beta$=0.3385, p=1.72×10$^{-2}$) and citations ($\beta$=0.5117, p=1.51×10$^{-2}$). Moreover, the search for non-academic jobs ($\beta$=−0.3022, p=3.21×10$^{-2}$) and the number of years on the job market ($\beta$=−0.3226, p=3.32×10$^{-2}$) were significantly negatively associated with offer status.

• Supplementary file 26. Applicants by their application type (R1 Universities, PUIs or both) and gender. Overview of job application survey respondents' (total and by gender) applications to R1 Universities (high-activity Research Universities), PUIs (Primarily Undergraduate Institutions; see *1* for definitions) or applied to both types of institutions. Percentages are calculated out of the total number of respondents to this particular survey question.

• Supplementary file 27. Teaching experience. Overview of the teaching experience (Teaching Assistantship for a course (lecture-based and/or laboratory-based) for the course instructor only versus beyond teaching assistantship which is independently designing and instructing undergraduate and/or graduate courses) of our applicant survey respondents. Percentages are calculated out of the total number of respondents to this particular survey question.

• Supplementary file 28. Themes from responses to question about teaching experiences beyond being a teaching assistant. Overview of specific types of teaching experience of our job applicant survey respondents detailed in a comment question. The "Adjunct Teaching Instructor for Undergraduate Courses at a

Community College or PUI" and "Adjunct Teaching Instructor for Undergraduate Courses at an R1 or PU Institution" were explicitly mentioned in comments by our applicant survey respondents. The "Total Adjunct teaching positions" were the total head-count of "adjunct type" college teaching performed by our job applicant survey respondents. A total of n = 162 applicants responded to this comment type long answer question.

• Supplementary file 29. Frequency of job applicant comments who received an offer. Overview of candidates who commented on their view in general of the application process. Responses to "Do you have any comments that you would like to share? For example, how did you experience the application process?" Survey participants were able to provide a long answer to this comment question. A word cloud referring to this table of comments is provided in *Figure 8C*. Percentages are calculated out of the total number of respondents to this particular survey question.

• Supplementary file 30. Applicant demographics: number of times (cycles/years) that the candidates had applied for faculty positions. Overview of number of times job candidate survey respondents applied for a faculty (PI) position (*Box 1*). This is in response to the survey question:"How many times have you applied for PI positions? i.e. if the 2018–2019 cycle was the first time, please enter "1", if you also applied last cycle, enter "2", etc. Percentages are calculated out of the total number of respondents to this particular survey question (n = 314).

• Supplementary file 31. General perceptions of the application process. Overview of candidates who commented on their view in general of the application process. Responses to "Do you have any comments that you would like to share? For example, how did you experience the application process?" Survey participants were able to provide long answers to this comment question. A word cloud referring to this table of comments is provided in *Figure 8C*.

• Supplementary file 32. Search committee survey: other comments. Overview of search committee members who commented on "Do you have any other comments or thoughts about the state of hiring for tenure track positions?" Survey participants were able to provide a long answer to this comment question.

• Supplementary file 33. Search committee survey: statistics. Overview of the search committee survey responses to "Approximately how many applicants for a posted position do you get?", "Approximately how many applicants make it through the first round of cuts?", "Approximately how many applicants are invited for off-site interview (Skype/phone)?", "Approximately how many offers does your committee make per job posting?", "Approximately how many openings has your department had in the last five years?", "Approximately how many applicants are invited for on-site interview?", "How long have you been involved in academic search committees?".

• Supplementary file 34. Search committee survey: demographics. Overview of the search committee faculty demographics of our faculty survey respondents. Percentages are calculated out of the total number of respondents to this particular survey question (n = 15).

• Supplementary file 35. Search committee survey: preprints. Overview of the search committee survey responses to "Does your committee look favorably upon preprints?".

• Supplementary file 36. Search committee survey: perceptions of the job market. Overview of the search committee survey responses to "What is your perception of the job market for tenure track faculty as someone involved in the search process (please tick all that are true)". Percentages are calculated out of the total number of respondents to this particular survey questions (*Rockey, 2012*).

• Supplementary file 37. Search committee survey: weighting given to various aspects of an application. Overview of the search committee survey responses to evaluation of a number of the tenure-track application materials: 1) "To what extent does the research proposal weigh on the selection process (e.g. "This candidate's research statement is incredibly compelling!", 2) "To what extent does good mentorship in the candidate's postdoctoral/graduate student lab explicitly weigh on selection process (e.g. "This candidate's mentor is known to produce good trainees", 3) "How heavily does the committee weigh graduate student fellowships or awards (e.g. The National Science Foundation (NSF) Graduate Research Fellowship (GRF), The National Institutes of Health (NIH) predoctoral fellowship/The Ruth L. Kirschstein National Research Service Awards for Individual Predoctoral Fellowships (F30 or F31), etc.)", 4) "How heavily does the committee weigh non-transitional postdoctoral fellowships or awards (e.g. NIH F32, AHA etc.)", 5) "Does your committee weigh Cell, Science, or Nature papers above papers in other journals?", 6) "To what extent does journal impact factor explicitly weigh in to the selection process (e.g. does the word 'impact factor' come up in discussions around applicants)?", 7)"How heavily does the committee weigh transition awards as a positive factor (i.e. The NIH Pathway to Independence (K99/R00) award, Burroughs Wellcome Career Award, or another award that provides the applicant with money as a new faculty member)?", 8)"How heavily does the committee weigh prior teaching experience?". In the survey, a 5-level Likert scale was used to record faculty impressions where a response of 1 = not at all and 5 = heavily. Percentages are calculated out of the total number of respondents to this particular survey question (n = 15).

• Supplementary file 38. Search committee survey: responses to the question "What information do you wish more candidates knew when they submitted their application?". Overview of the search committee who responded to "What information do you wish more candidates knew when they submitted their application?" Survey participants were able to provide a long

answer to this comment question. A word cloud referring to this table of comments is provided in *Figure 10A*.

• Supplementary file 39. Search committee survey: changes in the search process. Overview of search committee faculty members who commented on "Have you noticed any changes in the search process since the first search you were involved in?" Survey participants were able to provide a long answer to this question. A word cloud referring to this table of comments is provided in *Figure 10B*.

• Supplementary file 40. Applicant survey: scholarly metrics by gender with breakdown by offer status. Mean and median values for publication-related metrics plotted in *Figure 2B* broken down by gender and offer status. Additionally, p-values from Wilcoxon rank-sum tests that compare metric values from the female and male groups. "All" shows these values when the full dataset is considered, "With offers" shows values for only those applicants with at least one offer, and "Without offers" shows values for only those without any offers. "F" stands for female and "M" stands for male. Trends in gender differences remain the same even for the applicants with offers, serving as a possible explanation for the similar search outcomes for females and males and the importance of gender in the logistic regression.

• Supplementary file 41. The job applicant survey. Survey of the applicants to the tenure-track jobs.

• Supplementary file 42. The search committee survey. Survey of faculty members involved in tenure-track searches.

• Transparent reporting form

## Data availability

The authors confirm that, for approved reasons, access restrictions apply to the data underlying the findings. Raw data underlying this study cannot be made publicly available in order to safeguard participant anonymity and that of their organizations. Ethical approval for the project was granted on the basis that only aggregated data is provided (as has been provided in the supplementary tables) (with appropriate anonymization) as part of this publication.

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
