## [Decision Letter]

Your article has been reviewed by three peer reviewers, and the evaluation has been overseen by two members of the *eLife* Features Team. The following individuals involved in review of your submission have agreed to reveal their identity: Adriana Bankston (Reviewer #2).

Summary:

This article (which is based on a survey of more than 300 early-career researchers who are/were on the job market) has the potential to be important. However, the data require further analysis and the presentation and discussion need to be improved.

Essential revisions:

1. The authors should remove the Twitter poll and related analysis from this paper, leaving just the main survey and the faculty survey.

2. Throughout the manuscript, the authors seek to minimize the importance of papers published in Cell, Nature or Science (CNS). However, there are at least two instances in which having a CNS paper appears to be the strongest predictor of job market success (Table S7 and Figure 4B). Additionally, prior studies have also suggested the same (see Pinheiro et al., 2014; and links to multiple peer reviewed studies in Cuff, 2017). It is absolutely critical that the instances where having a CNS paper appears to be the strongest predictor of job market success be noted and described in the text. (See comments from reviewer #1 for a fuller discussion of this issue).

- Cuff, A. J. (2017). An Academic Lottery or a Meritocracy? Inside HigerEd. Retrieved from https://www.insidehighered.com/advice/2017/05/03/phds-need-real-data-how-potential-employers-make-hiring-decisions-essay.

- Pinheiro, D., Melkers, J., & Youtie, J. (2014). Learning to play the game: Student publishing as an indicator of future scholarly success. Technological Forecasting & Social Change, 81, 56-66. doi:10.1016/j.techfore.2012.09.008

3. Men have more first author publications than women particularly in CNS journals - the authors should comment on this.

4. In Figure 4D the authors should identify the factors that differentiate between candidates with one or more offers vs. those who received none.

5. Regarding the logistic regression: rather than conducting an analysis with 16 variables, many of which are collinear, the authors should conduct this analysis one variable at a time to identify the single features that predict job market success. The authors can then test these variables for collinearity and combine uncorrelated variables into models that include two, three, four, etc. independent predictors.

6. 96% of the respondents with CNS publications were in fields in "Biomedical or life sciences" or "Biology (other)" as described in Figure 1A. The authors should therefore repeat several of the key analyses on success predictors on the life science cohort alone, including making a new version of Figure 4 that includes data for the life science cohort alone. The authors should also update the tables S7, S9, S10 and S22 to include statistics about CNS publications done for the life scientists in the cohort alone (alongside the data already presented for the full cohort).

7. The authors conducted a "PUI-only" subgroup analysis (Figure 6), but it would also be informative to conduct an analysis among candidates who only applied to research-intensive positions. The differences between the positions may be noteworthy, and by combining too many unrelated job searches into one analysis, the authors may be missing out on important relationships.

8. The authors should provide some more background on the survey and how it was distributed at the start of the results section. What checks were in place to ensure that only early-career scientists answered it? Similarly, the authors should provide more information in the results section on how the search committee survey was distributed and who answered it.

9. This study really does not represent a cross section of different types of early career researchers from different fields nor "a wide variety of fields." Nor is it really an international applicant pool since data on race/ethnicity or nationality or citizenship status was not asked. It really represents a sample of postdocs (96%) from the biomedical and biological sciences with (72%) currently working within the U.S., Canada and U.K. who are on the job market. This is an important distinction to make because prior research Cantwell (2011), Cantwell & Taylor (2015), and Sauerman & Roach (2016) and others have done research in this area and shown how international postdocs working in the U.S. have had limited success in transitioning into tenure-track faculty positions and provide reasons to suggest why. The authors should discuss this at an appropriate place in the text and consider citing some or all of the following references:

- Cantwell, B., & Taylor, B. J. (2015). Rise of the science and engineering postdoctorate and the restructuring of academic research. Journal of Higher Education, 86(5), p 667-696.

- Cantwell, B. (2011). Transnational mobility and international academic employment: gatekeeping in an academic competition arena. Minerva: A Review of Science, Learning and Policy, 49(4), p 425-445.

- Sauermann, H. & Roach, M. (2016). Why pursue the postdoc path? Science 352:663-664. Doi: 10.1126/science.aaf2061

10. Another concern is an attempt to make correlations between number of applications with both number of interviews and offers, without taking into consideration the quality of the application, nor where the applicants received their training. There have been some large quantitative studies done by others (e.g. Clauset, Arbesman, & Larremore, 2015) that found faculty hiring follows a common and steeply hierarchical structure where doctoral prestige and where an applicant did both their Ph.D. and postdoc appointment better predicts hiring, especially in R1 institutions. The authors should discuss this concern at an appropriate place in the text and consider citing the following reference:

- Clauset, A., Arbesman, S., & Larremore, D. B. (2015). Systematic inequality and hierarchy in faculty hiring networks. Science Advances 1: e1400005. doi:10.1126/sciadv.1400005

11. Under "Statement of Ethics", the authors write that an IRB exemption was obtained on 08/29/2019. However, the survey itself was conducted in April 2019. I believe that retroactive IRB approval is generally prohibited by federal regulations. Was the IRB aware that the study had already been conducted when they granted the exemption? Can the authors comment on this serious discrepancy?

12. Teaching experience was not assessed either from the applicant or the institution perspective. Applicants self-disclosed their presumed teaching experience and then when a number of them did get job offers at R1s, the statement was made that applicants fulfill the teaching requirements for any university type. That is an oversimplification of findings. The authors' data indicates that the majority of applicants applied to R1s. This should be further discussed by the authors.

13. The authors could comment on whether the number of postdoc positions attained correlated with their ability to obtain faculty positions (in other words, are 3 postdoc appointments more desirable as compared to 1 when applying for a faculty position at a prestigious institution?).

14. The authors could address how the application process might be improved so it is a more positive experience.

---

## [Author Response]

[We repeat the reviewers’ points here in italic, and include our replies in Roman.]

Essential revisions:1. The authors should remove the Twitter poll and related analysis from this paper, leaving just the main survey and the faculty survey.

We have now removed the section below from the “Applicants perceive the process to be time-consuming and opaque, with minimal to no feedback” results, including the corresponding tables:

“A separate Twitter poll indicated that applicants in general, not specifically for this cycle, typically spend more than 3 hours tailoring each application (49) (Table S26 Supplementary File 1). Our pooled applicants at minimum then spent a combined 22,932 hours (7,644 applications x 3 hours preparation each), or 2.62 years, on these applications. Individually, this number amounts to 72 hours for each applicant on average, but does not take into account how long the initial creation of “base” application materials takes, which is often a much longer process. In another follow-up Twitter poll, a majority of respondents felt that time spent on preparing faculty job applications impeded their ability to push other aspects of their career forward (Table S27 Supplementary File 1) (50).”

2. Throughout the manuscript, the authors seek to minimize the importance of papers published in Cell, Nature or Science (CNS). However, there are at least two instances in which having a CNS paper appears to be the strongest predictor of job market success (Table S7 and Figure 4B). Additionally, prior studies have also suggested the same (see Pinheiro et al., 2014; and links to multiple peer reviewed studies in Cuff, 2017). It is absolutely critical that the instances where having a CNS paper appears to be the strongest predictor of job market success be noted and described in the text. (See comments from reviewer #1 for a fuller discussion of this issue).- Cuff, A. J. (2017). An Academic Lottery or a Meritocracy? Inside HigerEd. Retrieved from https://www.insidehighered.com/advice/2017/05/03/phds-need-real-data-how-potential-employers-make-hiring-decisions-essay- Pinheiro, D., Melkers, J., & Youtie, J. (2014). Learning to play the game: Student publishing as an indicator of future scholarly success. Technological Forecasting & Social Change, 81, 56-66. doi:10.1016/j.techfore.2012.09.008

We thank the reviewers for this comment, and would like to clarify our position. We agree with the reviewers that the discussion of the importance of CNS publications is a sensitive and well-known issue. The relationship between job market success and CNS publications is complex. We have read the references suggested by the reviewers, and we agree that several studies have concluded that CNS publications are generally held in high regard by search committees. Moreover, we discuss in the text that there is a high perceived importance of CNS papers to applicants. However, our data shows 2 clear findings with regards to CNS: 1) Most applicants who get offers do *not* have a CNS paper and 2) Applicants with a CNS paper appear to have a higher chance of getting an offer than those who do not have a CNS paper. We believe that the demonstration of the first point (most candidates do not have a CNS paper) is an important point for postdocs to consider but does not detract from the 2nd point (CNS papers appear to confer an advantage to an applicant). We note that an analysis of the offer association with CNS publications is shown in Figure 4D. We observed a 7-15% increase in offer rate for candidates which had published in CNS journals. We also note that despite this observation, ~60% of our survey respondents still received at least one offer. Moreover, our data (Figure 4C) suggest that those individuals with CNS authorship who also received a job offer were more highly cited than those with CNS authorship without a job offer, suggesting that assessment of the impact of an applicant’s work is a complex blend of many factors.

Our small survey of faculty search committee members, revealed an attitude towards CNS publications that was discordant with that of the applicants and the conventional expectation. While we acknowledge that we do not have extensive information on the decision making process of every single search committee, the volume of current literature around this issue speaks to the culture shift that is occurring around academic hiring. Formal discussions of CNS as a metric and its impact on equity in the professoriate have resulted in the development of guidelines such as DORA, that reflect the burgeoning culture change. While an in-depth analysis of impact is outside the scope of this work, we look forward to future work on this topic.

3. Men have more first author publications than women particularly in CNS journals - the authors should comment on this.

We note that these results are presented in Figure 2B and 2C, as well as discussed in the “applicant scholarly metrics by gender” results subsection. How these results are in line with previous findings is presented within the introductory paragraph to that results section. We additionally revisit this point in the discussion stating “Further, despite the fact that women face numerous challenges in academia, including underrepresentation at the faculty level in most STEM departments (52,53,56,57), and trail men in publication-related metrics (Figure 2B), our data suggest very few differences in outcomes in the May 2018-May 2019 female applicant pool relative to their male counterparts. Both genders received similar numbers of interviews and offers, and gender-based differences in publication-related metrics persisted even when considering only the 185 individuals with offers, suggesting that committees are becoming increasingly aware of gender bias in publication-related metrics and are taking them into account when evaluating applicants (Table S40 Supplementary File 1). We have also now added a further point in the discussion pertaining specifically to the intersection of gender and CNS publications. “First, examination of our data revealed a gender gap in publication metrics, with males reporting more CNS authorship and publications overall, indicating that opportunities for publication are not equally available (52,53). ” Our survey has captured numerous aspects of the academic job market and the factors influencing the success of applicants. We recognize that gender (and other marginalized identities) play a large role. We believe the role of gender is thoroughly discussed in our manuscript while allowing in depth analysis of other factors.

4. In Figure 4D the authors should identify the factors that differentiate between candidates with one or more offers vs. those who received none.

We thank the reviewers for this comment, and for the opportunity to clarify this point within our manuscript. In our study, we performed two different analyses to understand which factors differentiate candidates based on their success in obtaining offers. We performed Wilcoxon tests (presented in Figure 4B and Table S7 Supplementary File 1) comparing the offer percentage (the number of offers/the number of applications) and also performed a logistic regression comparing candidates with offers vs those without (Table S22). The data from these two analyses are summarized in Figure 4D, as noted by the reviewer, but were confusing due to the wording in both the text and the figures. We have clarified this in the text by adding further details regarding the offer status and positive correlations in the following sentence: “When missing values were imputed, significant positive coefficients were observed for having a higher h-index, higher application numbers, career transition awards and identifying as female and obtaining an offer.” Further, we have clarified this in the figure legend for Figure 4D as follows: “Summary of significant results testing criteria associated with 1+ offer (positive) or no offers (negative) with offer outcomes either through Wilcoxon analyses (Table S7 Supplementary File 1) or logistic regression (Table S24 Supplementary File 1) ordered by decreasing effect size.”

5. Regarding the logistic regression: rather than conducting an analysis with 16 variables, many of which are collinear, the authors should conduct this analysis one variable at a time to identify the single features that predict job market success. The authors can then test these variables for collinearity and combine uncorrelated variables into models that include two, three, four, etc. independent predictors.

We thank the reviewer for bringing this to our attention. As suggested by the reviewer, we have undertaken a greedy stepwise variable selection procedure in which we started with the variable that was the most predictive of offer status (1+ offer vs 0 offers) and then incrementally added one variable at a time, selecting for the best pair, triplet, and so on. While doing this we also ensured that no new variable was highly correlated or anticorrelated with a previously included variable (Absolute Spearman correlation coefficient cutoff of 0.5). This resulted in the exclusion of two variables that were highly correlated with citation count: h-index and total number of publications (see correlation plot below).

**Author response image 1. sa2fig1:** 

Among the remaining 14 variables, the combination of seven variables resulted in the highest accuracy in cross-validation experiments: (Applying to) Other jobs, Application number, Citations, Years on job market, Postdoc fellowships, (transition to independence) Funding and CNS (co-authorship) (see Author response table 1 below).Author response table 1.

We note that while this 7-variable model is more accurate in predicting offer status than the full model presented in the earlier version, there is a possibility of bias typically arising from the subjective decisions made during a variable selection exercise. For instance, the choice of a greedy stepwise approach, the use of Spearman correlation as a measure of collinearity, the correlation coefficient thresholds, the choice of accuracy as the quantity to maximize, among others, are all likely to influence which variables are selected and how many are deemed optimal. Nonetheless, the agreement between the single-variable analyses (Wilcoxon tests) and the more rigorous logistic regression analysis gives us confidence to update the logistic regression analysis in the manuscript to this 7-variables model (see updated Figures 4 and Table S24 Supplementary File 1). We have also modified the text summarizing these results in the “Interplay between metrics” section to read as follows:

"We implemented a rigorous variable selection procedure to maximize accuracy and remove highly correlated variables. This resulted in a model that included only seven variables (Table S24 Supplementary File 1) that was tested on a subset of applicants (n=105) who provided answers across all variables. This regression model revealed that a higher number of applications, a higher citation count and obtaining a postdoctoral fellowship were significantly associated with receipt of an offer. When missing values were imputed and the full applicant pool (n=317) was considered, all previous variables remained significant, and a significant positive coefficient was also observed for having a career transition award. In both versions of the model, the search for non-academic jobs was significantly negatively associated with offer status (Figure 4D). We note that the model with imputed data was more accurate than that with missing values excluded at distinguishing between applicants with and without offers in 10-fold cross-validation experiments. However this accuracy was found to only be 69.6%, which is insufficient to construct a usable classifier of offer status. Due to the predominance of applicants from the life sciences in our dataset, we also repeated these analyses on a subset containing only these applicants. While more variables were included in the model, the general trends remained the same, with the addition of the number of years spent on the job market as a significant negative factor in receiving an offer (Table S25 Supplementary File 1; Figure 4 – Supplement 1)."

6. 96% of the respondents with CNS publications were in fields in "Biomedical or life sciences" or "Biology (other)" as described in Figure 1A. The authors should therefore repeat several of the key analyses on success predictors on the life science cohort alone, including making a new version of Figure 4 that includes data for the life science cohort alone. The authors should also update the tables S7, S9, S10 and S22 to include statistics about CNS publications done for the life scientists in the cohort alone (alongside the data already presented for the full cohort).

We have updated Table S7 in Supplementary File 1 and added new tables S9 and S11 equivalents for Life/Biomedical Sciences applicants (respondents who indicated their field of research as Chemistry, Biology, Bioengineering or Biomedical or Life Sciences) scholarly metrics to the Supplementary file. The new added tables are numbered S10 and S12 in Supplementary File 1. Figure 4 – Supplement 1 shows the same analyses as in Figure 4, but restricted to the life sciences cohort. Table S25 in Supplementary File 1 reports the coefficients, p-values and other related information in the same manner as in Table S24 in Supplementary File 1.

7. The authors conducted a "PUI-only" subgroup analysis (Figure 6), but it would also be informative to conduct an analysis among candidates who only applied to research-intensive positions. The differences between the positions may be noteworthy, and by combining too many unrelated job searches into one analysis, the authors may be missing out on important relationships.

We have updated Figure 6 to include comparisons of candidates who only applied to PUIs (PUI-focused), those who only applied to R1 institutions (R1 Focused) and candidates who applied to both. Our original observations still hold (candidates applying to PUI positions are more likely to have more extensive teaching experience). However the new analysis also reveals some interesting trends (the R1 Focused subgroup is majority male in contrast to the two other groups) that we now comment on in the “Research versus Teaching-intensive institutions” section.

8. The authors should provide some more background on the survey and how it was distributed at the start of the results section. What checks were in place to ensure that only early-career scientists answered it? Similarly, the authors should provide more information in the results section on how the search committee survey was distributed and who answered it.

We have altered the text in two separate places to provide clarity on this point. The first paragraph under “Academic Job Applicant Demographics” has been changed to include statements that participants in the survey self-identified as applicants in the academic job market in 2018-2019. The first paragraph under “Search Committees Value the Future” has been changed to state that the search committee survey was sent to a limited number of faculty, from within the authors’ professional networks, who were known to serve on search committees.

9. This study really does not represent a cross section of different types of early career researchers from different fields nor "a wide variety of fields." Nor is it really an international applicant pool since data on race/ethnicity or nationality or citizenship status was not asked. It really represents a sample of postdocs (96%) from the biomedical and biological sciences with (72%) currently working within the U.S., Canada and U.K. who are on the job market. This is an important distinction to make because prior research Cantwell (2011), Cantwell & Taylor (2015), and Sauerman & Roach (2016) and others have done research in this area and shown how international postdocs working in the U.S. have had limited success in transitioning into tenure-track faculty positions and provide reasons to suggest why. The authors should discuss this at an appropriate place in the text and consider citing some or all of the following references:- Cantwell, B., & Taylor, B. J. (2015). Rise of the science and engineering postdoctorate and the restructuring of academic research. Journal of Higher Education, 86(5), p 667-696.- Cantwell, B. (2011). Transnational mobility and international academic employment: gatekeeping in an academic competition arena. Minerva: A Review of Science, Learning and Policy, 49(4), p 425-445.- Sauermann, H. & Roach, M. (2016). Why pursue the postdoc path? Science 352:663-664. Doi: 10.1126/science.aaf2061

We thank the reviewer for this comment and note that the complete demographic data of survey participants' country of origin, field of scientific research, and current position is included in Figure 1 and detailed in Tables S2, S3 and S5 in Supplementary File 1. We acknowledge that our survey did not fully capture the difficulties faced by non-citizens in obtaining a US faculty position, though these difficulties have been demonstrated in previous literature. We have added that citizenship status was a common issue thought to hinder applicants’ progress in the last line of the last paragraph of the “Applicants perceive the process to be time-consuming and opaque, with minimal to no feedback” section and highlight its prominence in Figure 7B. We have also added the following text to the discussion of the limitations of our study: “As indicated in our open question responses (Figure 7B), international postdocs may be specifically challenged in obtaining faculty job offers in the United States and Europe due to immigration policies as well as how mobility is interpreted by the job market (59).” We note that we had previously referenced Sauermann, H. & Roach, M. (2016) in our introduction, it is reference number 9.

10. Another concern is an attempt to make correlations between the number of applications with both number of interviews and offers, without taking into consideration the quality of the application, nor where the applicants received their training. There have been some large quantitative studies done by others (e.g. Clauset, Arbesman, & Larremore, 2015) that found faculty hiring follows a common and steeply hierarchical structure where doctoral prestige and where an applicant did both their Ph.D. and postdoc appointment better predicts hiring, especially in R1 institutions. The authors should discuss this concern at an appropriate place in the text and consider citing the following reference:- Clauset, A., Arbesman, S., & Larremore, D. B. (2015). Systematic inequality and hierarchy in faculty hiring networks. Science Advances 1:e1400005. doi:10.1126/sciadv.1400005

While we acknowledge that variables such as training institution and prestige of the applicants’ graduate school and postdoctoral mentors undoubtedly influence who obtains faculty job offers, studying this relationship was beyond the scope of the current study. The current study was not designed to explore these variables because of privacy concerns for our respondents. We did not inquire about respondents’ training institutions, advisors’ prestige, or other metrics associated with their “networks” which may have influenced their job search success. Future work to explore these variables are certainly needed. We have cited Clauset et al in our discussion section when we discuss the potential impact of these training/mentoring/network variables we did not assess in the current study. From “Challenges in the Academic Job Market” in the Discussion:

“Additionally, other unmeasured factors (e.g. applicant pedigree) are likely important considerations, consistent with recent data implicating institutional prestige and non-meritocratic factors in faculty hiring (51). This should be a major consideration for future studies of the academic job market.”

We would like to add that our open-ended question regarding “What was helpful for your application” (Figure 7A) indicated applicants perceived networking methods and pedigree to be valuable in helping them land faculty jobs. So, there is clearly merit to study these metrics in future work.

11. Under "Statement of Ethics", the authors write that an IRB exemption was obtained on 08/29/2019. However, the survey itself was conducted in April 2019. I believe that retroactive IRB approval is generally prohibited by federal regulations. Was the IRB aware that the study had already been conducted when they granted the exemption? Can the authors comment on this serious discrepancy?

We initially conducted the surveys for the Future PI Slack community only. When we decided on data analysis and wider dissemination of the results, we applied for IRB exemption for use of existing data. We have now corrected our statement of ethics to reflect the exemption criteria for publication of the survey data: “The surveys used in this manuscript were designed and implemented by the authors listed above on a voluntary basis outside of their post-doctoral positions at the time. The authors respect the confidentiality and anonymity of all respondents, and no identifiable private information was collected. Participation in both surveys was voluntary and the respondents could stop responding to the surveys at any time. The use of the data collected in these surveys was determined to meet the exemption criteria for secondary use of existing data [45 CRF 46.104 (d)(4)] by the University of North Dakota Institutional Review Board (IRB) on 08/29/2019. IRB project number: IRB-201908-045. Please contact Dr. Amanda Haage (amanda.haage@und.edu) for further inquiries.”

12. Teaching experience was not assessed either from the applicant or the institution perspective. Applicants self-disclosed their presumed teaching experience and then when a number of them did get job offers at R1s, the statement was made that applicants fulfill the teaching requirements for any university type. That is an oversimplification of findings. The authors' data indicates that the majority of applicants applied to R1s. This should be further discussed by the authors.

We have revised the teaching section within the results section to address the reviewers’ comments about oversimplification of the findings. The statement that applicants fulfill the teaching requirement for any university type has been removed, as requested by reviewers. Below is the revised text.

**“**Levels of teaching experience varied among respondents

Discussion surrounding the academic job market is often centered on applicants' publications and/or funding record, while teaching experience generally receives much less attention. Accordingly, a candidate’s expected teaching credentials and experience vary, largely depending on the type of hiring institution. We asked applicants whether they focused their applications to a specific type of institution (R1, PUI, or both; see Box 1 for definitions), allowing us to examine teaching experience across R1 and/or PUI applicants. Most respondents applied to jobs at R1 institutions (Figure 5A), which may explain the focus on research-centric qualifications. It remains unclear what the emphasis on teaching experience is for search committees at R1 institutions, however the literature suggests that there seems to be a minimal focus (47). Additionally, there might be differences in departmental or institutional requirements that are unknown to outsiders. What is commonly accepted is that many applications to an R1 institution requires a teaching philosophy statement. The majority (99%) of our survey respondents have teaching experience (Figure 5B), with roughly half of applicants’ experience limited to serving as a Teaching Assistant (TA) (Box 1), and half reporting experience beyond a TA position, such as serving as an instructor of record (Figure 5B). The degree of teaching experience did not change based on the target institution of the applicant (Figure 5C), nor did the percentage of offers received significantly differ between groups based on teaching experience (Figure 5D).”

13. The authors could comment on whether the number of postdoc positions attained correlated with their ability to obtain faculty positions (in other words, are 3 postdoc appointments more desirable as compared to 1 when applying for a faculty position at a prestigious institution?).

We did not look for a correlation between the number of postdoc positions and the ability to obtain faculty positions since the length of postdoctoral positions vary widely in our dataset (i.e. 1 postdoctoral position does not necessarily infer a shorter length of training compared to training spanning across more than one postdoctoral positions) and the majority of our respondents had a single postdoctoral appointment (see Figure 1E). In our survey, we did ask respondents for the total number of years (across all appointments) of postdoc training (Figure 1D) and this further reveals field-specific differences in expectations of postdoc appointments and length. Therefore we believe that the relationship between likelihood of receiving a faculty job offer and training extent is complex and likely requires additional considerations (e.g. quality of training over a training timespan, length of graduate training, etc) and is beyond the scope of the analysis in this paper.

14. The authors could address how the application process might be improved so it is a more positive experience.

We note the reviewer’s desire to provide recommendations for hiring institutions and applicants going on the job market. We have added a paragraph to the conclusion of the paper to address how the application process might be improved, so that it is a more positive experience. Additionally, we plan to share our perspective and recommendations for policy changes in a companion piece currently under preparation that provides opinion on data reported here.

The following text has been added to the conclusion of the manuscript:

**“**Conclusions

The faculty job search process lacks transparency and data regarding what makes a successful applicant. Here, we began to address this need via a job market survey targeted towards the applicants themselves, including their perceptions of the application process. Of over 300 responses by job applicants, we did not receive a single positive comment on the process, despite the fact that 58% of our applicants received at least one job offer. Our data suggest that baseline thresholds exist for those more likely to receive a faculty job offer, but that many different paths can lead to a job offer. This variety of paths likely reflects both the applicant's preparation as well as different evaluation criteria used by individual search committees. For these reasons, we urge applicants not to conclude that lower than average metrics in any one area are automatically disqualifying. Indeed, we believe that increasing the transparency of the application process through systematic data collection will allow a more detailed study of the many paths to obtaining a faculty offer.

Our data show that there is a mental strain on applicants during this process, and we propose a number of potential solutions with the understanding that faculty hiring is a complex process involving multiple stakeholders. We believe the application process could be improved by simplifying the process, including standardizing application materials (e.g. requirements for research statements are similar for R1 institutions) and requesting references only after candidates are shortlisted, so that the burden of application preparation time can be reduced. Constructive feedback from mentors is vital for success during the application and interview preparation stages. Additionally, if possible, communication from search committees about unsuccessful applications would be helpful. We understand that these points may increase the workload of mentors and search committees but, if put into place, could alleviate some of the stress related to the job application process. In addition, applicants need to work to be sure their materials are strong and well-researched as the quality of these materials and demonstrating fit for a job posting are important to faculty on search committees (47). More work is needed to understand the challenges search committees face in order to improve their experience of the application process.

It is our hope that this and future work will not only allow all stakeholders to make informed decisions, but will also enable critical examination, discussion, and reassessment of the implicit and explicit values and biases being used to select the next generation of academic faculty. Such discussions are crucial in building an academic environment that values and supports all of its members.”